# Performance Evaluation for Launcher Testing Vehicle

Teodor-Viorel Chelaru [ID], Valentin Pană * and Costin Ene

Faculty of Aerospace Engineering, Univesity Politehnica of Bucharest, 060042 Bucharest, Romania
* Correspondence: valentin.pana@upb.ro; Tel.: +40-771-096917

**Abstract:** The paper's purpose is to present a calculus model for a testing vehicle that can be used to validate guidance, navigation and control systems for reusable launchers in all flight phases. The technical solution is based on a throttleable engine with thrust vectoring control and a reaction control system (RCS) used for roll. For calculus, we will develop a nonlinear model with six degrees of freedom, based on quaternion, extended with nonlinear equations that use pulse modulation in order to control roll. In order to synthesize the controller, we also develop a linear model similar to the launcher model. The paper analyzes two basic scenarios, first with the ascending and the descending flight phases and the second having a horizontal flight interleaved between ascending and descending flight phases, both scenarios being specific for reusable launchers. Based on these scenarios, the paper evaluates some performances of the proposed vehicle, namely flight envelope and guidance accuracy.

**Keywords:** mathematical model; guidance navigation and control system; reaction control system; testing vehicle; reusable launcher; flight envelope; guidance accuracy



## 1. Introduction

In the field of commercial launchers, nowadays, there are two directions for reducing the costs: achievement of the launcher families with modular launchers like ANGARA [1] and DELTA [2], which allow insertion on different orbits of all sorts of payloads and achievement of the reusable launchers, as it is done in the FALCON program [3,4], developed by Space X. A problem stemming from these trends is the development of an advanced navigation, control and guidance system that can solve the guidance problem for the entire launcher family or make complex maneuvers for the reusable launchers. A Launcher Testing Vehicle (LTV) can be used as an independent stage to solve these complex problems with a low development cost, with the propulsion and control system to test in flight and the launcher guidance navigation and control system (GNC).

In our vision, the Launch Testing Vehicle (LTV) is similar to the vehicle used for landing on a celestial body (Moon, Mars). Discussing technical solutions, the reference landing vehicle Apollo Lunar Module [5,6] uses a descending throttleable engine for landing and an ascent engine for docking. This approach also uses a reaction control system (RCS) having two blocks with four thrusters each for attitude control. An alternative technical solution for a Lunar Lander Demonstrator (LLD) is presented in the paper [7]. LLD has five large descending thrusters and eight small thrusters for attitude control. Many fixed thrusters allow the use of on/off modulation to control roll pitch and yaw without throttling or gimbaling for thrust vectoring control (TVC).

Among the important projects for reusable launchers, we can mention STARSHIP [8], developed by SpaceX Company(Hawthorne, USA), whose technical solution is based on the multiple Raptor Engine with gimbal capability and aerodynamic surfaces (flaps) for control. Another is the CALLISTO project [9], developed in cooperation with Centre National d'Etude Spatiales (CNES), Japan Aerospace Exploration Agency (JAXA), and Deutsches Zentrum fur Luft-und Raumfahrt (DLR), whose technical solution is based on the RV-X engine, developed by JAXA, which will have gimbaling, throttling and re-ignition

capability. Moreover, the project supposes the aerodynamic surfaces (fins) and RCS with 6–8 thrusters for control.

Different from these solutions, specific for landing vehicles, in this paper, we will consider a Launch Testing Vehicle (LTV) based on thrust vectoring control (TVC) using a throttleable engine with gimbal capability for pitch and yaw, and separately, two RCS blocks with two thrusters each with Pulse Width -Pulse Frequency modulation (PWPF) [10,11] for roll control. In order to simplify the technical solution, we do not use aerodynamic surfaces, which are inefficient for low velocity.

Currently, a number of current studies [12,13] focus on determining an optimal evolution, in terms of low fuel consumption, by applying convex optimization methods on models with three degrees of freedom (3DOF) for complex evolutions.

Unlike these, the current paper proposes developing a calculation model with six degrees of freedom (6DOF), similar to that of the launchers, to underpin the vehicle's subsequent development and conduct preliminary flight tests. The model will focus on the movement's control and stability, aiming to evaluate the vehicle's performance in terms of the flight envelope and guidance accuracy. For the synthesis of the guidance system, linear models for longitudinal and lateral movement in the local frame based on the Hamilton quaternion are developed. Applying Linear Quadratic Regulator (LQR) synthesis, we will obtain an optimal regulator, and the gains used in control signals will be defined.

For performance evaluation, two simple hypothetical scenarios will be considered, one for vertical evolution, which allows the determination of the maximum altitude, and one for horizontal evolution, which allows the determination of the maximum flight distance, both cases for a fixed amount of fuel. Based on these two scenarios, for two test cases, the landing accuracy will be evaluated in the conditions of the parametric uncertainty of the model as well as the noise introduced by the sensors. Additionally, for a test case in the second scenario, the uniform and turbulent wind influence on lateral movement will be evaluated.

Taking into consideration the work objectives to implement a calculus model for LTV, which will be used to validate GNC for the launcher, the reference frames used are similar to those used for the launcher model described in detail in prior work [14,15]. This is characterized by the y-axis of the local frame oriented vertically upwards and, in the case of Euler angles, the order of rotations 3-2-1 from the local frame to the body frame.

Similar to launchers, because of their shape, the LTV has the aerodynamic focus in front of the center of mass, translating into static instability, which leads to a similar approach regarding the control problem. Different from the launcher, for LTV, some simplifying assumptions related to low velocities and distance domains will be made concerning the frames and the movement equations.

In summary, the novelties proposed by the paper, related to other similar approaches, are:

A simplified technical control solution based only on propulsion. The technical solution is similar to lunar landing vehicles. Still, it is customized for the evolution of launchers by considering the aerodynamic terms in the model and the atmospheric, disturbing effects.

Formulation of the control problem based on linear coordinates in the local frame and the quaternion. Unlike our previous paper [16], we introduced the control of linear coordinates to improve guidance, especially landing accuracy, and we used quaternions to ensure the robustness of the model

Evaluation of system performance in the flight envelope and landing accuracy with a highlight on the influence of uniform wind, turbulent wind, sensor noise, and uncertainty of the model parameters.

The paper's major contribution, which can be considered a contribution to the state of the art, consists of base movement analysis and linear model, both dedicated for the testing vehicle.

As will be shown during the work, for the technical solution adopted, the basic movement is close to landing on a celestial body despite the presence of the atmosphere.

Regarding the linear form of the motion equations, unlike most works [17] that use the velocity frame for the linear form of translational equations, the local frame will be used in our work, which will facilitate the control of the linear coordinates. Appendix A of the paper will present the method of obtaining the coupled linear form of the equations of motion.

In addition, for the avoidance of singularities and the system's robustness for describing the attitude, including in the linear form, the quaternion was used, which constitutes an additional element regarding the contributions of the work.

For the verification of the model but also for the design of a methodology for evaluating the vehicle's performance, two flight scenarios will be considered in both situations. The test was performed with the control based on the previously specified linear model, which was included in the 6 DOF nonlinear model.

## 2. Motion Equations

### 2.1. Reference Frames, Rotation Matrix

As we mentioned earlier, the reference frames used will be similar to those used for the micro launcher model. Because LTV has a limited range, up to 10 km around the launch position, some simplifying assumptions will be added to the launcher model. So, we will consider the Earth flat, without rotation; hence we will use only the two frames described below for the equation of motion.

(a)    *The Start frame/ The Local frame—$OX_0Y_0Z_0$*

This coordinate system denoted $OX_0Y_0Z_0$ Figure 1 has its origin on the Earth's surface at the start point of the vehicle. Axis $Y_0$ is oriented vertically upwards. The frame has $X_0$ axis orientated to the launch direction. $Z_0$ axis completes the right frame to the right of the launch plane. This coordinate system is considered to be an inertial reference frame.

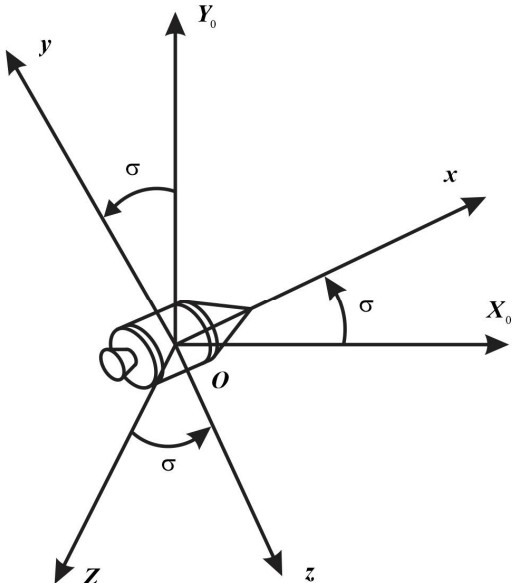

**Figure 1.** The link between local frame and body frame.

(b)    *The Body frame—$Oxyz$*

This coordinate system originates in the center of mass of the vehicle Figure 1. The axis $x$ is along the longitudinal symmetry axis of the body. Axis $y$ is in a symmetry plane of the vehicle. The axis $z$ arises, forming with the first two axes a right trihedral. Next, we use this trihedral to write dynamic rotation equations around the mass center. Also, this will be used to write thrust and aerodynamic terms.

The rotation between the start frame $OX_0Y_0Z_0$ and body frame $Oxyz$ can be obtained using Euler's angles in order 3-2-1 or using Rodrigues parameters, or using quaternion

components, as we can see from Figure 1. Next, for robustness, we will use quaternion components. First, we recall the definition of quaternion components:

$$q_1 = l \sin \frac{\sigma}{2}; \ q_2 = m \sin \frac{\sigma}{2}; \ q_3 = n \sin \frac{\sigma}{2}; \ q_4 = \cos \frac{\sigma}{2} \tag{1}$$

where $l$, $m$, $n$ are the projection of the unit vector of the rotation axis:

$$\boldsymbol{e}_\sigma = \boldsymbol{I}l + \boldsymbol{J}m + \boldsymbol{K}n \tag{2}$$

and $\sigma$ are the principal rotation angle.

The total rotation matrix from the local frame to the body frame according to the paper [10] is:

$$\boldsymbol{A}_I = \begin{bmatrix} q_4^2 + q_1^2 - q_2^2 - q_3^2 & 2(q_1 q_2 + q_3 q_4) & 2(q_3 q_1 - q_2 q_4) \\ 2(q_1 q_2 - q_3 q_4) & q_4^2 + q_2^2 - q_3^2 - q_1^2 & 2(q_2 q_3 + q_4 q_1) \\ 2(q_3 q_1 + q_2 q_4) & 2(q_2 q_3 - q_4 q_1) & q_4^2 + q_3^2 - q_1^2 - q_2^2 \end{bmatrix}. \tag{3}$$

*2.2. Translation Equation in Local Frame*

Considering the start frame as inertial, applying the impulse theorem, we obtain:

$$\dot{\boldsymbol{V}}_0 = \boldsymbol{B}_I m^{-1} (\boldsymbol{F} + \boldsymbol{T}) + \boldsymbol{g}_0 , \tag{4}$$

where:

$\boldsymbol{F} = X^A \boldsymbol{i} + Y^A \boldsymbol{j} + Z^A \boldsymbol{k}$—Aerodynamic force in body frame;
$\boldsymbol{T} = X^T \boldsymbol{i} + Y^T \boldsymbol{j} + Z^T \boldsymbol{k}$—Thrust force in body frame;
$\boldsymbol{g}_0$—Gravitational acceleration

and $\boldsymbol{B}_I$ matrix is the transpose of the rotation matrix $\boldsymbol{A}_I$, (3):

$$\boldsymbol{B}_I = \begin{bmatrix} q_4^2 + q_1^2 - q_2^2 - q_3^2 & 2(q_1 q_2 - q_3 q_4) & 2(q_3 q_1 + q_2 q_4) \\ 2(q_1 q_2 + q_3 q_4) & q_4^2 + q_2^2 - q_3^2 - q_1^2 & 2(q_2 q_3 - q_4 q_1) \\ 2(q_3 q_1 - q_2 q_4) & 2(q_2 q_3 + q_4 q_1) & q_4^2 + q_3^2 - q_1^2 - q_2^2 \end{bmatrix}. \tag{5}$$

To the previous equation, we add the translational kinematic equations:

$$\begin{bmatrix} \dot{x}_0 & \dot{y}_0 & \dot{z}_0 \end{bmatrix}^T = \begin{bmatrix} V_x & V_y & V_z \end{bmatrix}^T. \tag{6}$$

Regarding mass variation in time, this is due to the fuel consumption:

$$\dot{m}(t) = -T_0 \delta_T I_{sp}^{-1}, \tag{7}$$

where $I_{sp}$ is the specific impulse of the rocket engine, $T_0$ is maximum thrust and $\delta_T$ is throttling command.

*2.3. Rotational Equations in Body Frame*

To obtain rotational dynamic equations, we start with the angular momentum theorem. As we know, the angular momentum theorem shows that a moment applied to a body will change the angular momentum. In this case, taking into consideration that the body frame is a non-inertial frame, we can write:

$$\dot{\boldsymbol{h}} = \boldsymbol{\Omega} \times \boldsymbol{h} = \boldsymbol{M}, \tag{8}$$

where: $\boldsymbol{h}$—body angular momentum; $\boldsymbol{M}$—applied moment; $\boldsymbol{\Omega}$—angular velocity of the body frame relative to the inertial frame (start frame/local frame).

The relation can be written in matrix form:

$$\begin{bmatrix} \dot{p} \\ \dot{q} \\ \dot{r} \end{bmatrix} = J^{-1} \begin{bmatrix} L \\ M \\ N \end{bmatrix} - J^{-1} A_\Omega J \begin{bmatrix} p \\ q \\ r \end{bmatrix}, \tag{9}$$

where $A_\Omega$ is the antisymmetric matrix associated with angular velocity vector $\mathbf{\Omega}$:

$$A_\Omega = \begin{bmatrix} 0 & -r & q \\ r & 0 & -p \\ -q & p & 0 \end{bmatrix}. \tag{10}$$

$J$ represents the inertial momentum matrix. If the additional hypothesis is made that the vehicle has two symmetry planes, the inertial matrix becomes:

$$J = \begin{bmatrix} A & 0 & 0 \\ 0 & B & 0 \\ 0 & 0 & C \end{bmatrix}, \tag{11}$$

where:

$$A = \int \left( y^2 + z^2 \right) dm; B = \int \left( z^2 + x^2 \right) dm; C = \int \left( x^2 + y^2 \right) dm, \tag{12}$$

$L, M, N$ are the components of the applied moment along the body frame axis given by aerodynamics terms and thrust terms:

$$L = L^A + L^T; \; M = M^A + M^T; \; N = N^A + N^T, \tag{13}$$

and:

$$\mathbf{\Omega} = \begin{bmatrix} p & q & r \end{bmatrix}^T, \tag{14}$$

denotes the rotation velocity of the body frame relative to the inertial frame expressed by projections along the body frame.

Equation (9) can be written in scalar form:

$$\begin{aligned} \dot{p} &= [L + qr(B - C)]A^{-1}; \\ \dot{q} &= [M + rp(C - A)]B^{-1}; \\ \dot{r} &= [N + pq(A - B)]C^{-1}. \end{aligned} \tag{15}$$

According to paper [10], the associated kinematic equations allow obtaining the quaternion components:

$$\begin{bmatrix} \dot{q}_1 \\ \dot{q}_2 \\ \dot{q}_3 \\ \dot{q}_4 \end{bmatrix} = \frac{1}{2} \begin{bmatrix} q_4 & -q_3 & q_2 \\ q_3 & q_4 & -q_1 \\ -q_2 & q_1 & q_4 \\ -q_1 & -q_2 & -q_3 \end{bmatrix} \begin{bmatrix} p \\ q \\ r \end{bmatrix}. \tag{16}$$

### 2.4. Guidance Signals

As we mentioned at the beginning of the paper, because LTV has the purpose of testing the launcher GNC, its structure will be similar to that described in [18–20]. The main control signals result from the quaternion error between the current and reference values. Different from the usual control system of a launcher for LTV, we will use error signals between linear coordinates.

Considering the desired attitude quaternion: $\begin{bmatrix} q_{1d} & q_{2d} & q_{3d} & q_{4d} \end{bmatrix}^T$, and the current attitude quaternion $\begin{bmatrix} q_1 & q_2 & q_3 & q_4 \end{bmatrix}^T$, the control error quaternion $\begin{bmatrix} q_{1e} & q_{2e} & q_{3e} & q_{4e} \end{bmatrix}^T$ can be obtained by using the relation indicated in papers [7,10]:

$$\begin{bmatrix} q_{1e} \\ q_{2e} \\ q_{3e} \\ q_{4e} \end{bmatrix} = \begin{bmatrix} q_{4d} & q_{3d} & -q_{2d} & -q_{1d} \\ -q_{3d} & q_{4d} & q_{1d} & -q_{2d} \\ q_{2d} & -q_{1d} & q_{4d} & -q_{3d} \\ q_{1d} & q_{2d} & q_{3d} & q_{4d} \end{bmatrix} \begin{bmatrix} q_1 \\ q_2 \\ q_3 \\ q_4 \end{bmatrix}. \tag{17}$$

With this error quaternion, as is shown in work [10], we can obtain the guidance signal for LTV evolution:

$$\begin{aligned} u_l &= -2k_2 q_{1e} - k_1 p \\ u_m &= -2k_6 q_{2e} - k_5 q + u_x a_{3,1} + u_y a_{3,2} + u_z a_{3,3} \\ u_n &= u_{n0} - 2k_4 q_{3e} - k_3 r - u_x a_{2,1} - u_y a_{2,2} - u_z a_{2,3} \\ u_T &= u_{To} - u_u - u_x a_{1,1} - u_y a_{1,2} - u_z a_{1,3} \end{aligned}, \tag{18}$$

where $a_{i,j}$ are the elements of the rotation matrix $A_I$ (3).

The signal for velocity control along the body axis which will contribute to throttling command:

$$u_u = k_7(u - u_d) = k_7 \left[ (V_x - V_{xd})a_{1,1} + (V_y - V_{yd})a_{1,2} + (V_z - V_{zd})a_{1,3} \right]. \tag{19}$$

In relation to (18), we denoted $u_{To}$; $u_{no}$ the basic signals corresponding to equilibrium evolution.

The control signals for linear coordinate along the local frame axis $u_x$, $u_y$, $u_z$ will be defined next, separately for ascending/descending evolution and for horizontal evolution.

(a)    *Control signals for linear coordinate during ascending/descending evolution*

For ascending /descending evolutions, besides the control signals previously defined, we will use the control signals for linear coordinates along the local frame axis:

$$u_x = k_8 x_0 + k_9 V_x; \; u_y = k_9 \left( V_y - V_{yd} \right); \; u_z = k_8 z_0 + k_9 V_z. \tag{20}$$

(b)    *Control signals for linear coordinate during horizontal evolution*

If we desire to have a horizontal evolution, the control signal for error quaternion and axial velocity is supplemented by a control signal for linear coordinates in the local frame:

$$u_x = k_9(V_x - V_{xd}); \; u_y = k_{10}(y_0 - y_{0d}) + k_9 V_y; \; u_z = k_8 z_0 + k_9 V_z. \tag{21}$$

*2.5. Aerodynamic Terms*

Due to large incidence angles ($\pm 180°$), determination of aerodynamics terms for the vehicle is difficult. On the other hand, aerodynamical terms are important because they allow the introduction of statical instability of the vehicle due to the position of aerodynamic focus related to the center of mass. Moreover, we will analyze wind influence on vehicle flight through aerodynamic terms and define the base movement for horizontal flight.

In order to solve this problem, we can use the approach proposed in the CALLISTO project [21] by experimental activities in Wind Tunnel. This requires an advanced stage of the project when the vehicle configuration is established. In our preliminary study, we must do some simplified calculus for aerodynamics terms when the vehicle configuration is not precisely definite. Unfortunately, for slender body shapes, the theoretical results are valid only in the case of a small incidence angle [22]. In this case, we will make the following simplifying hypothesis, the vehicle has a spherical shape, and the position of the aerodynamic focus is in front of the mass center.

Based on this hypothesis, the components of the aerodynamic force in the body frame will be considered as drag force opposite to velocity components for each axis:

$$X^A = S \frac{\rho V^2}{2} C_X; \; Y^A = S \frac{\rho V^2}{2} C_Y; \; Z^A = S \frac{\rho V^2}{2} C_Z, \tag{22}$$

where, due to symmetry, will have:

$$C_X = -C_d\frac{u}{V}; C_Y = -C_d\frac{v}{V}; C_Z = -C_d\frac{w}{V}, \tag{23}$$

where $C_d$ is the drag coefficient for a sphere that depends on the Reynolds number, see Figure 2, according to paper [23].

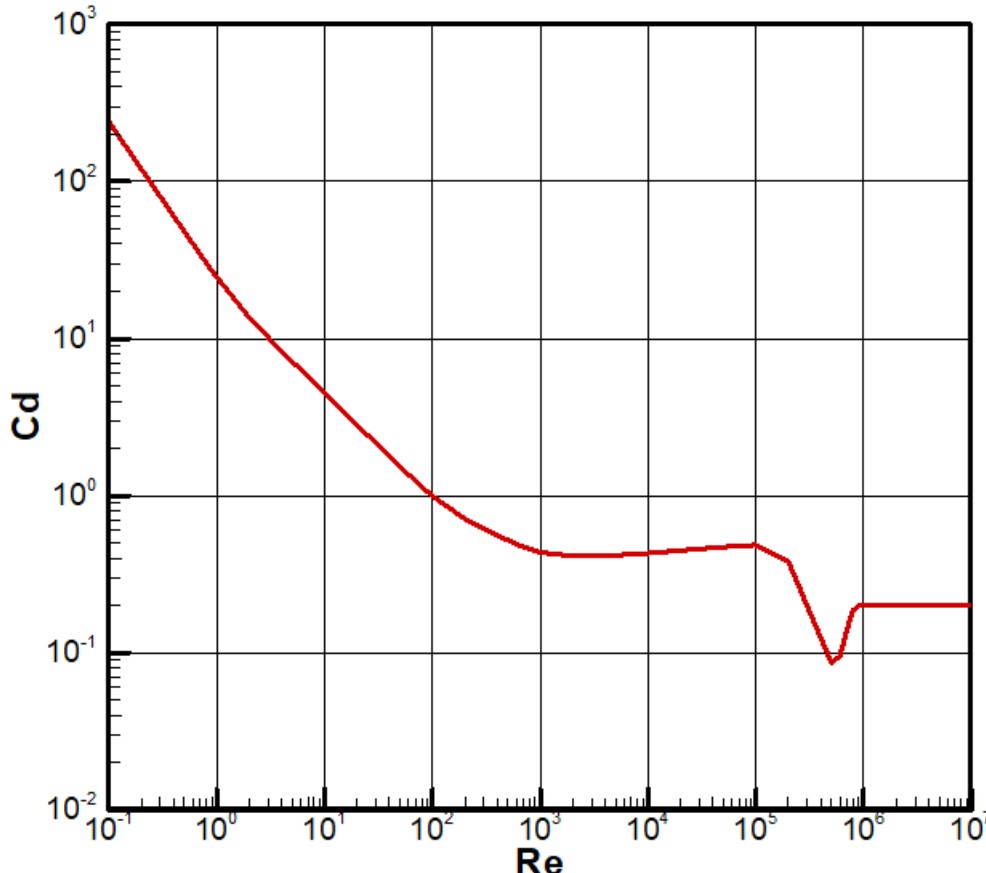

**Figure 2.** Drag coefficient for the sphere depending on Reynolds number.

Considering aerodynamic focus placed at a distance denoted with $x_F$ in front of the mass center, the aerodynamic moments are:

$$L^A = 0, \ M^A = -x_F Z^A, N^A = x_F Y^A. \tag{24}$$

For LTV configuration, similar to the case of launchers $x_F > 0$.

*2.6. Thrust Vectoring Control*

The guidance signals previously obtained (18) are applied to the thrust vectoring control system (TVC), which can be approximated with a first-order delay element:

$$\dot{\delta}_l\tau_l = -\delta_l + u_l; \ \dot{\delta}_m\tau_m = -\delta_m + u_m; \ \dot{\delta}_n\tau_n = -\delta_n + u_n; \ \dot{\delta}_T\tau_T = -\delta_T + u_T, \tag{25}$$

which provides command terms: angular deflections or linear displacements, which are used in the thrust terms described next.

Similar to the launcher, the thrust terms are given by thrust vector projection along the body frame, namely by a nozzle or entire engine tilt [24,25]. If we use two tilting angles $\delta_n$, $\delta_m$ similar to the launcher, for pitch and yaw, we can write:

$$X^T = \delta_T T_0 cos\delta_m cos\delta_n;$$
$$Y^T = -\delta_T T_0 cos\delta_m sin\delta_n; \quad (26)$$
$$Z^T = \delta_T T_0 sin\delta_m,$$

where $T_0$ is the maximum thrust of the engine and $\delta_T$ is the throttling command with a unitary maximum value. Considering the position of the thrust vector application at the exit of the engine nozzle, we can write the moment commands:

$$M^T = -x_T Z^T; \ N^T = x_T Y^T, \quad (27)$$

where:

$$x_T = x_{cm} - l, \quad (28)$$

in which $x_{cm}$ is the current position of the center of the mass from the top of the LTV, and $l$ is the reference length (the distance between the body top and nozzle exit section).

Regarding the roll command, we suppose the existence of a Reaction Control System (RCS), which ensures the roll control through two RCS blocks with two thrusters on each one with monopropellant cold gas or another equivalent propeller, as is shown in [25].

### 2.7. Reaction Control System in Roll

In order to form the roll command, we start from the roll guidance signal $u_l$. This signal is applied to the actuator, which in this case is RCS.

As we can see from Figure 3, there are two blocks of the reactive elements located in the *yoz* plane of the LTV body, which allows obtaining the roll command with the reactive elements $(\delta_1, \delta_2, \delta_3; \delta_4)$.

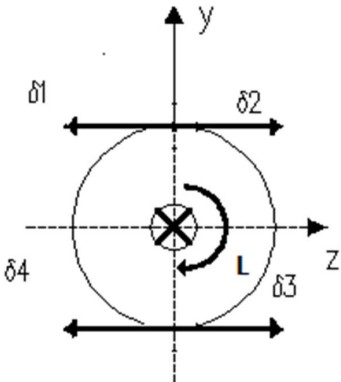

**Figure 3.** Schematic diagram of the four reactive elements—RCS.

To formulate the problem, we will consider the $\delta_i$ as oriented vectors along nozzles (Figure 3) with the module in binary form, having a value "1" if the reactive element is working or "0" if the reactive element is not working.

As shown in [10,11,26], the system uses Pulse Width-Pulse Frequency modulation (PWPF) for roll control (Figure 4). PWPF comprises a linear integrator that allows a system supplementary adjustment and a nonlinear element (N), Type Schmidt Trigger [11,25]. To follow the input, a feedback reaction loop is used.

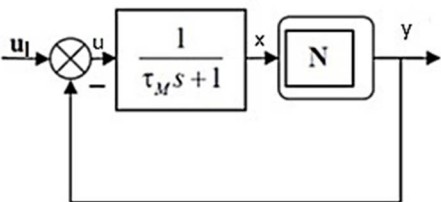

**Figure 4.** PWPF system for roll control.

The nonlinear element which switches between reactive elements can be equivalent to the relay type with hysteresis and dead zone, as shown in Figure 5.

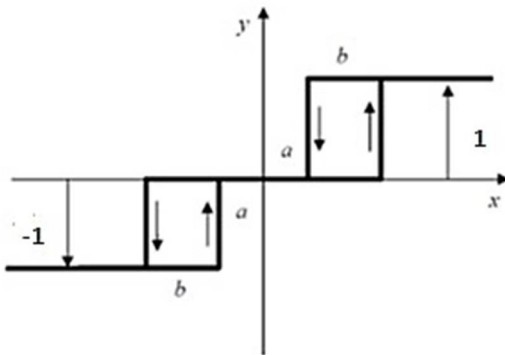

**Figure 5.** Operating scheme for nonlinear element (N).

As shown in Figure 5, the dimension of the dead zone is $2a$, hysteresis width is $b$, and the output in saturation is $\pm 1$.

For passing from the continuous command $u_l$ to the binary values of the elements $\delta_i$ we first form the intermediate signals:

$$u = u_l - y. \tag{29}$$

The signal $u$ is integrated, obtaining the signal $x$:

$$\dot{x}\tau_M = -x + u. \tag{30}$$

The signal $x$, is passed through the nonlinear element shown in Figure 5, obtaining signal $y$

$$y = N(x), \tag{31}$$

where we denoted with $N(x)$ the nonlinear element applied to $x$ signal in the open loop:

$$
\begin{aligned}
y = 1 &\rightarrow (\delta_1 = 0; \delta_2 = 1; \delta_3 = 0; \delta_4 = 1) \\
y = -1 &\rightarrow (\delta_1 = 1; \delta_2 = 0; \delta_3 = 1; \delta_4 = 0). \\
y = 0 &\rightarrow (\delta_1 = 0; \delta_2 = 0; \delta_3 = 0; \delta_4 = 0)
\end{aligned}
\tag{32}
$$

Using the reactive elements, we obtain the roll equivalent command:

$$\delta_l = \frac{\delta_2 + \delta_4 - \delta_1 - \delta_3}{2}. \tag{33}$$

We can observe that, due to the binary structure of the elements $\delta_i$, the equivalent roll command $\delta_l$ can have values "$\pm 1$" or "0".

Finally, using the Fourier transform [27], we can obtain a linear differential equation:

$$\dot{\delta}_l \tau_l \cong -\delta_l + u_l \tag{34}$$

similar to relation (25).

Given the previous reactive force of the RCS element $R$ and body diameter $d$, we can obtain the roll command moment given by:

$$L^T = Rd\delta_l. \tag{35}$$

## 3. Base Movement, Linear Form of the Equation of Motion, Stability and Command Matrices

To define the base movement, we start from the dynamic equations for translation (4) and rotation (9). Due to the initial hypothesis for the equation of motion, the Earth is

considered a flat surface that does not rotate; the gravity acceleration does not depend on the orientation of the local frame, which can be arbitrarily chosen. In this case, we can suppose, without reducing the problem generality, that the base movement takes place in a vertical plane $X_0OY_0$ of a local frame, the lateral motion being null. Simultaneously, we suppose a configuration with two symmetry planes adopted for the inertia matrix in dynamic rotation equations, and we can choose the roll angle null. That means in the base movement; we have only $q_3$ as an independent parameter, the other quaternions components being:

$$q_1 = 0; q_2 = 0; q_4 = \left(1 - q_3^2\right)^{1/2}. \tag{36}$$

In this case, the rotation matrix (5) becomes:

$$\boldsymbol{B}_I = \begin{bmatrix} 1 - 2q_3^2 & -2q_3q_4 & 0 \\ 2q_3q_4 & 1 - 2q_3^2 & 0 \\ 0 & 0 & 1 \end{bmatrix}. \tag{37}$$

From the translation equation, we will consider the relations in the vertical plane:

$$\begin{aligned} m\dot{V}_x &= \left(X^A + X^T\right)\left(1 - 2q_3^2\right) - 2\left(Y^A + Y^T\right)q_3q_4 \\ m\dot{V}_y &= 2\left(X^A + X^T\right)q_3q_4 + \left(Y^A + Y^T\right)\left(1 - 2q_3^2\right) - G, \end{aligned} \tag{38}$$

where we denoted $G = mg$, and from rotation equations, we hold the pitch equation:

$$\dot{r}C = N^T + N^A, \tag{39}$$

If we cancel the velocity derivatives, it results:

$$\begin{aligned} \left(X^A + X^T\right)\left(1 - 2q_3^2\right) - 2\left(Y^A + Y^T\right)q_3q_4 &= 0; \\ 2\left(X^A + X^T\right)q_3q_4 + \left(Y^A + Y^T\right)\left(1 - 2q_3^2\right) - G &= 0, \end{aligned} \tag{40}$$

and from the pitch equation, we cancel the angular acceleration, obtaining:

$$N^T + N^A = 0. \tag{41}$$

The nonlinear system (40), (41) can be solved with Newton's iterative algorithm. If we impose as a base movement the horizontal flight at a specified altitude and velocity, from Equations (40) and (41), we obtain the component $q_3$, and thrust components $X^T$, $Y^T$, the thrust moment in pitch $N^T$ being linked to the component $Y^T$ of the thrust given by Equation (27).

From thrust components, we can obtain the throttling command $\delta_T$ and the angular pitch thrust command $\delta_n$, which means the equilibrium commands:

$$\delta_T = T_0^{-1}\sqrt{\left(X^T\right)^2 + \left(Y^T\right)^2}; \ \delta_n = -\text{atan}\frac{Y^T}{X^T}. \tag{42}$$

Taking into consideration that the actuators in Equation (25) have unitary gain, the guidance signals equilibrium commands $u_{T0}$ and $u_{n0}$ from the relation (18) can be obtained with the relations (42).

Having base movement defined, considering as commands: throttling command $\delta_T$ the angular deflections $\delta_m$, $\delta_n$, and equivalent command $\delta_l$ the stability matrix and the command matrix can be obtained. For cross-checking, the stability and command matrices were determined both numerically and by analytical relations, obtaining identical results. For clarity, next, we present their analytical expressions.

Because in the base movement, the longitudinal motion is decupled from lateral motion, we will separately analyze these two motions.

Starting from the coupled linear development presented in Appendix A, for longitudinal motion, having the states $Vx, Vy, r, q_3, x, y$ and the commands $\delta_T$, $\delta_n$, the linear form of the equations are:

$$\Delta \dot{V}_x = -\frac{KV}{m} \quad \left(1 + \frac{aV_x^2}{V^2}\right)\Delta V_x - \frac{aK}{m}\frac{V_x V_y}{V}\Delta V_y + \frac{2\delta_T T_0}{q_4 m}\left[(1 - 2q_3^2)sin\delta_n - 2q_3 q_4 cos\delta_n\right]\Delta q_3$$
$$-\frac{bK}{m}VV_x\Delta y_0 + \frac{T_0}{m}\left[2q_3 q_4 sin\delta_n + (1 - 2q_3^2)cos\delta_n\right]\Delta\delta_T$$
$$+\frac{T_0\delta_T}{m}\left[2q_3 q_4 cos\delta_n - (1 - 2q_3^2)sin\delta_n\right]\Delta\delta_n ;$$

$$\Delta \dot{V}_y = -\frac{aK}{m} \quad \frac{V_x V_y}{V}\Delta V_x - \frac{KV}{m}\left(1 + \frac{aV_y^2}{V^2}\right)\Delta V_y + \frac{2\delta_T T_0}{q_4 m}\left[(1 - 2q_3^2)cos\delta_n + 2q_3 q_4 sin\delta_n\right]\Delta q_3$$
$$-\frac{bK}{m}VV_y\Delta y_0 + \frac{T_0}{m}\left[2q_3 q_4 cos\delta_n - (1 - 2q_3^2)sin\delta_n\right]\Delta\delta_T$$
$$-\frac{T_0\delta_T}{m}\left[2q_3 q_4 sin\delta_n + (1 - 2q_3^2)cos\delta_n\right]\Delta\delta_n ;$$

$$\Delta \dot{r} = \frac{KV x_F}{C} \quad \left[2q_3 q_4\left(1 + \frac{aV_x^2}{V^2}\right) - (1 - 2q_3^2)\frac{aV_x V_y}{V^2}\right]\Delta V_x \tag{43}$$
$$+\frac{KV x_F}{C}\left[2q_3 q_4\frac{aV_x V_y}{V^2} - (1 - 2q_3^2)\left(1 + \frac{aV_y^2}{V^2}\right)\right]\Delta V_y$$
$$+2\frac{KV x_F}{q_4 C}\left[(1 - 2q_3^2)V_x + 2q_3 q_4 V_y\right]\Delta q_3$$
$$-\frac{bK}{C}x_F V\left[(1 - 2q_3^2)V_y - 2q_3 q_4 V_x\right]\Delta y_0 - \frac{x_T T_0}{C}sin\delta_n\Delta\delta_T$$
$$-\frac{x_T T_0\delta_T}{C}cos\delta_n\Delta\delta_n$$

$$\Delta \dot{q}_3 = \frac{q_4}{2}\Delta r ;$$
$$\Delta \dot{x}_0 = \Delta V_x ;$$
$$\Delta \dot{y}_0 = \Delta V_y ;$$

where $K = \frac{S}{2}\rho C_d$ and the drag dependence of the velocity and the altitude is given by:

$$a = 1 + V\frac{\partial C_d}{C_d\partial V} \cong 1. - 0.5683{\cdot}V + 0.6409 \times 10^{-1}{\cdot}V^2 - 0.2332 \times 10^{-2}{\cdot}V^3;$$

$$b = \frac{\partial C_d}{C_d\partial y_0} + \frac{\partial\rho}{\rho\partial y_0} \cong 0.318 \times 10^{-4} .$$

Denoting:

$$a_{Vx}^{Vx} = -\frac{KV}{m}\left(1 + \frac{aV_x^2}{V^2}\right); \; a_{Vx}^{Vy} = -\frac{aK}{m}\frac{V_x V_y}{V}; a_{Vx}^{q3} = \frac{2\delta_T T_0}{q_4 m}\left[(1 - 2q_3^2)sin\delta_n - 2q_3 q_4 cos\delta_n\right];$$

$$a_{Vx}^{y} = -\frac{bK}{m}VV_x;$$

$$a_{Vy}^{Vx} = -\frac{aK}{m}\frac{V_x V_y}{V}; a_{Vy}^{Vy} = -\frac{KV}{m}\left(1 + \frac{aV_y^2}{V^2}\right); \; a_{Vy}^{q3} = \frac{2\delta_T T_0}{q_4 m}\left[(1 - 2q_3^2)cos\delta_n + 2q_3 q_4 sin\delta_n\right];$$

$$a_{Vy}^{y} = -\frac{bK}{m}VV_y;$$

$$a_r^{Vx} = \frac{KV x_F}{C}\left[2q_3 q_4\left(1 + \frac{aV_x^2}{V^2}\right) - (1 - 2q_3^2)\frac{aV_x V_y}{V^2}\right];$$

$$a_r^{Vy} = \frac{KV x_F}{C}\left[2q_3 q_4\frac{aV_x V_y}{V^2} - (1 - 2q_3^2)\left(1 + \frac{aV_y^2}{V^2}\right)\right]; \; a_r^{q3} = 2\frac{KV x_F}{q_4 C}\left[(1 - 2q_3^2)V_x + 2q_3 q_4 V_y\right];$$

$$\tag{44}$$

$$a_r^{y} = -\frac{bK}{C}x_F V\left[(1 - 2q_3^2)V_y - 2q_3 q_4 V_x\right];$$

$$b_{Vx}^{\delta T} = \frac{T_0}{m}\left[2q_3 q_4 sin\delta_n + (1 - 2q_3^2)cos\delta_n\right]; \; b_{Vx}^{\delta n} = \frac{T_0\delta_T}{m}\left[2q_3 q_4 cos\delta_n - (1 - 2q_3^2)sin\delta_n\right];$$

$$b_{Vy}^{\delta T} = \frac{T_0}{m}\left[2q_3 q_4 cos\delta_n - (1 - 2q_3^2)sin\delta_n\right]; b_{Vy}^{\delta n} = -\frac{T_0\delta_T}{m}\left[2q_3 q_4 sin\delta_n + (1 - 2q_3^2)cos\delta_n\right];$$

$$b_r^{\delta T} = -\frac{x_T T_0}{C}sin\delta_n; \; b_r^{\delta n} = -\frac{x_T T_0\delta_T}{C}cos\delta_n,$$

we obtain for the longitudinal motion the stability matrix in Table 1

**Table 1.** Stability matrix for longitudinal motion *A*.

$$
\begin{bmatrix}
a_{Vx}^{Vx} & a_{Vx}^{Vy} & a_{Vx}^{q3} & a_{Vx}^{y} \\
a_{Vy}^{Vx} & a_{Vy}^{Vy} & a_{Vy}^{q3} & a_{Vy}^{y} \\
a_{r}^{Vx} & a_{r}^{Vy} & a_{r}^{q3} & a_{r}^{y} \\
& & \frac{q4}{2} & \\
1 & & & \\
& 1 & &
\end{bmatrix}
$$

and the command matrix in Table 2.

**Table 2.** Command matrix for longitudinal motion *B*.

$$
\begin{bmatrix}
b_{Vx}^{\delta T} & b_{Vx}^{\delta n} \\
b_{Vy}^{\delta T} & b_{Vy}^{\delta n} \\
b_{r}^{\delta T} & b_{r}^{\delta n} \\
& \\
&
\end{bmatrix}
$$

Because in longitudinal motion, the controlled states differ from the ascending/descending evolutions to the horizontal evolution, we will define a separate stability matrix and command matrix for each type of longitudinal evolution.

For longitudinal motion in ascending/descending evolution, the states are $Vx, Vy, r, q_3, x$, and the commands are $\delta_T, \delta_n$. Because in this particular case, the basic movement has supplementary conditions:

$$
Vx = 0; \ \delta_n = 0; \ q_3 = q_4 = \frac{\sqrt{2}}{2}; \ \left(1 - 2q_3^2\right) = 0, \ 2q_3 q_4 = 1, \tag{45}
$$

the stability matrix has the form from Table 3,

**Table 3.** Stability matrix for longitudinal motion in ascending/descending evolution *A*.

$$
\begin{bmatrix}
a_{Vx}^{Vx} & & a_{Vx}^{q3} & \\
& a_{Vy}^{Vy} & & \\
a_{r}^{Vx} & & a_{r}^{q3} & \\
& & \frac{\sqrt{2}}{4} & \\
1 & & &
\end{bmatrix}
$$

and the command matrix has the form from Table 4.

**Table 4.** Command matrix for longitudinal motion in ascending/descending evolution *B*.

$$
\begin{bmatrix}
& b_{Vx}^{\delta n} \\
b_{Vy}^{\delta T} & \\
& b_{r}^{\delta n} \\
& \\
&
\end{bmatrix}
$$

where:

$$
\begin{aligned}
& a_{Vx}^{Vx} = -\frac{K}{m}V; \ a_{Vx}^{q3} = -2\sqrt{2}\frac{\delta_T T_0}{m} \ ; \ a_{Vy}^{Vy} = -(1+a)\frac{K}{m}V; \\
& a_{r}^{Vx} = 2\frac{Kx_F}{C}V; a_{r}^{q3} = 2\sqrt{2}\frac{Kx_F}{C}VV_y; b_{Vx}^{\delta n} = \frac{T_0 \delta_T}{m}; b_{Vy}^{\delta T} = \frac{T_0}{m}; b_{r}^{\delta n} = -\frac{x_T T_0 \delta_T}{C}.
\end{aligned} \tag{46}
$$

For longitudinal motion in horizontal evolution, the states are $Vx, Vy, r, q_3, y$ and the commands are $\delta_T, \ \delta_n$. Because in this particular case, the basic movement has the supplementary condition:

$$V_y = 0, \tag{47}$$

The stability matrix has the form from Table 5,

**Table 5.** Stability matrix for longitudinal motion in horizontal evolution *A*.

$$\begin{bmatrix} a_{Vx}^{Vx} & & & a_{Vx}^{q3} & \\ & a_{Vy}^{Vy} & & & \\ a_r^{Vx} & & & a_r^{q3} & \\ & & & & \frac{\sqrt{2}}{4} \\ 1 & & & & \end{bmatrix}$$

and the command matrix has the form from Table 6.

**Table 6.** Command matrix for longitudinal motion in horizontal evolution *B*.

$$\begin{bmatrix} & b_{Vx}^{\delta n} \\ b_{Vy}^{\delta T} & \\ & b_r^{\delta n} \\ & \\ & \end{bmatrix}$$

where:

$$\begin{aligned}
&a_{Vx}^{Vx} = -(1+a)\frac{KV}{m}; \ a_{Vx}^{q3} = \frac{2\delta_T T_0}{q_4 m}\left[(1-2q_3^2)sin\delta_n - 2q_3q_4cos\delta_n\right]; \ a_{Vx}^y = -\frac{bK}{m}VV_x; \\
&a_{Vy}^{Vy} = -\frac{KV}{m}; \ a_{Vy}^{q3} = \frac{2\delta_T T_0}{q_4 m}\left[(1-2q_3^2)cos\delta_n + 2q_3q_4sin\delta_n\right]; \ a_r^{Vx} = (1+a)\frac{KVx_F}{C}2q_3q_4; \\
&a_r^{Vy} = -\frac{KVx_F}{C}(1-2q_3^2); \ a_r^{q3} = 2\frac{KVx_F}{q_4 C}\left[(1-2q_3^2)V_x\right]; a_r^y = \frac{bK}{C}x_F V2q_3q_4 V_x; \\
&b_{Vx}^{\delta T} = \frac{T_0}{m}\left[2q_3q_4sin\delta_n + (1-2q_3^2)cos\delta_n\right]; \ b_{Vx}^{\delta n} = \frac{T_0\delta_T}{m}\left[2q_3q_4cos\delta_n - (1-2q_3^2)sin\delta_n\right]; \\
&b_{Vy}^{\delta T} = \frac{T_0}{m}\left[2q_3q_4cos\delta_n - (1-2q_3^2)sin\delta_n\right]; b_{Vy}^{\delta n} = -\frac{T_0\delta_T}{m}\left[2q_3q_4sin\delta_n + (1-2q_3^2)cos\delta_n\right]; \\
&b_r^{\delta T} = -\frac{x_T T_0}{C}sin\delta_n; \ b_r^{\delta n} = -\frac{x_T T_0\delta_T}{C}cos\delta_n.
\end{aligned} \tag{48}$$

Starting from the coupled linear development presented in Appendix A, for lateral motion, the states are $V_z, \ p, \ q, q_1, q_2, z$ and the commands are $\delta_l, \ \delta_m$ the linear form of the equations are:

$$\Delta\dot{V}_z = -\frac{KV}{m}\Delta V_z + \frac{2\delta_T T_0}{m}(q_3cos\delta_n - q_4sin\delta_n)\Delta q_1 - \frac{2\delta_T T_0}{m}(q_4cos\delta_n + q_3sin\delta_n)\Delta q_2 +$$

$$\frac{T_0\delta_T}{m}\Delta\delta_m;$$

$$\Delta\dot{p} = \frac{Rd}{A}\Delta\delta_l;$$

$$\Delta\dot{q} = \frac{KVx_F}{B}\Delta V_z + 2\frac{KVx_F}{B}(q_3V_x - q_4V_y)\Delta q_1 + 2\frac{KVx_F}{B}(q_4V_x + q_3V_y)\Delta q_2 - \frac{x_T T_0\delta_T}{B}\Delta\delta_m; \tag{49}$$

$$\Delta\dot{q}_1 = \frac{q_4}{2}\Delta p - \frac{q_3}{2}\Delta q;$$

$$\Delta\dot{q}_2 = \frac{q_3}{2}\Delta p + \frac{q_4}{2}\Delta q;$$

$$\Delta\dot{z}_0 = \Delta V_z.$$

Denoting:

$$a_{Vz}^{Vz} = -\frac{KV}{m}; \; a_{Vz}^{q1} = \frac{2\delta_T T_0}{m}(q_3 cos\delta_n - q_4 sin\delta_n); \; a_{Vz}^{q2} = -\frac{2\delta_T T_0}{m}(q_4 cos\delta_n + q_3 sin\delta_n);$$

$$a_q^{Vz} = \frac{KV x_F}{B}; \; a_q^{q1} = 2\frac{KV x_F}{B}(q_3 V_x - q_4 V_y); \; a_q^{q2} = 2\frac{KV x_F}{B}(q_4 V_x + q_3 V_y) \tag{50}$$

$$b_{Vz}^{\delta m} = \frac{T_0 \delta_T}{m}; \; b_p^{\delta l} = \frac{Rd}{A}; \; b_q^{\delta m} = -\frac{x_T T_0 \delta_T}{B},$$

we obtain for the lateral motion the stability matrix shown in Table 7

**Table 7.** Stability matrix for lateral motion **A**.

$$\begin{bmatrix} a_{Vz}^{Vz} & & a_{Vz}^{q1} & a_{Vz}^{q2} \\ a_q^{Vz} & & a_q^{q1} & a_q^{q2} \\ & \frac{q_4}{2} & \frac{-q_3}{2} & \\ & \frac{q_3}{2} & \frac{q_4}{2} & \\ 1 & & & \end{bmatrix}$$

and the command matrix shown in Table 8.

**Table 8.** Command matrix for lateral motion **B**.

$$\begin{bmatrix} & b_{Vz}^{\delta m} \\ b_p^{\delta l} & \\ & b_q^{\delta m} \\ & \\ & \end{bmatrix}$$

For lateral motion in ascending/descending evolution, considering condition (45), the matrix elements become:

$$a_{Vz}^{Vz} = -\frac{K}{m}V; \; a_{Vz}^{q1} = \frac{\sqrt{2}\delta_T T_0}{m}; \; a_{Vz}^{q2} = -\frac{\sqrt{2}\delta_T T_0}{m};$$

$$a_q^{Vz} = \frac{K x_F}{B}V; \; a_q^{q1} = -\sqrt{2}\frac{K x_F}{B}V V_y; \; a_q^{q2} = \sqrt{2}\frac{K x_F}{B}V V_y; \tag{51}$$

$$b_{Vz}^{\delta m} = \frac{T_0 \delta_T}{m}; \; b_p^{\delta l} = \frac{Rd}{A}; \; b_q^{\delta m} = -\frac{x_T T_0 \delta_T}{B}; \; q_3 = q_4 = \frac{\sqrt{2}}{2}.$$

For lateral motion in horizontal evolution, considering condition (47), the matrix elements become:

$$a_{Vz}^{Vz} = -\frac{K}{m}V; \; a_{Vz}^{q1} = \frac{2\delta_T T_0}{m}(q_3 cos\delta_n - q_4 sin\delta_n); \; a_{Vz}^{q2} = -\frac{2\delta_T T_0}{m}(q_4 cos\delta_n + q_3 sin\delta_n);$$

$$a_q^{Vz} = \frac{K x_F}{B}V; \; a_q^{q1} = 2\frac{K x_F}{B}V V_x q_3; \; a_q^{q2} = 2\frac{K x_F}{B}V V_x q_4; \tag{52}$$

$$b_{Vz}^{\delta m} = \frac{T_0 \delta_T}{m}; \; b_p^{\delta l} = \frac{Rd}{A}; \; b_q^{\delta m} = -\frac{x_T T_0 \delta_T}{B}.$$

## 4. Linear Form of the Guidance Signals, the Regulator's Design

The linearization of guidance signals will be done related to the system's states:

Velocity in local frame: $V_x$; $V_y$; $V_z$; Coordinate in local frame: $x_0$; $y_o$; $z_0$; Angular velocity in body frame: $p$; $q$; $r$; Attitude (quaternion): $q_1$; $q_2$; $q_3$. First, we linearized the error quaternion given by relation (17). Considering for the basic movement, the desired quaternion is equal to the basic quaternion, the relation (17) becomes:

$$
\begin{bmatrix} \Delta q_{1e} \\ \Delta q_{2e} \\ \Delta q_{3e} \\ \Delta q_{4e} \end{bmatrix} = \begin{bmatrix} q_4 & q_3 & 0 & 0 \\ -q_3 & q_4 & 0 & 0 \\ 0 & 0 & q_4 & -q_3 \\ 0 & 0 & q_3 & q_4 \end{bmatrix} \begin{bmatrix} \Delta q_1 \\ \Delta q_2 \\ \Delta q_3 \\ \Delta q_4 \end{bmatrix} + \begin{bmatrix} \Delta q_{4d} & \Delta q_{3d} & -\Delta q_{2d} & -\Delta q_{1d} \\ -\Delta q_{3d} & \Delta q_{4d} & \Delta q_{1d} & -\Delta q_{2d} \\ \Delta q_{2d} & -\Delta q_{1d} & \Delta q_{4d} & -\Delta q_{3d} \\ \Delta q_1 d & \Delta q_{2d} & \Delta q_{3d} & \Delta q_{4d} \end{bmatrix} \begin{bmatrix} 0 \\ 0 \\ q_3 \\ q_4 \end{bmatrix}. \tag{53}
$$

Separating the deviation of the desired quaternion as an input signal and considering:

$$
\Delta q_4 = -\frac{q_3}{q_4} \Delta q_3.\ ;\ \Delta q_{4d} = -\frac{q_3}{q_4} \Delta q_{3d}, \tag{54}
$$

we obtain:

$$
\begin{aligned}
\Delta q_{1e} &= q_4 \Delta q_1 + q_3 \Delta q_2 + f_{1d}; \\
\Delta q_{2e} &= q_4 \Delta q_2 - q_3 \Delta q_1 + f_{2d}; \\
\Delta q_{3e} &= \frac{1 - 2q_3^2}{q_4} \Delta q_3 + f_{3d},
\end{aligned} \tag{55}
$$

where the input signals are:

$$
f_{1d} = -q_4 \Delta q_{1d} - q_3 \Delta q_{2d};\ f_{2d} = -q_4 \Delta q_{2d} + q_3 \Delta q_{1d};\ f_{3d} = -\frac{1 - 2q_3^2}{q_4} \Delta q_{3d}. \tag{56}
$$

Next, we linearized the axial velocity control (19). Considering the basic movement, the linear form of the relation (19) becomes:

$$
\Delta u_u = k_7 \Delta u + \Delta f_u = k_7 \left[ a_{1,1} \Delta V_x + a_{1,2} \Delta V_y + V_x \Delta a_{1,1} + V_y \Delta a_{1,2} \right] + \Delta f_u, \tag{57}
$$

where $\Delta f_u$ is an input signal containing desired axial velocities.

On the other hand, for basic movement, the elements of the rotation matrix (3) become:

$$
\begin{aligned}
a_{1,1} &= \left(1 - 2q_3^2\right);\ a_{1,2} = 2q_3 q_4\ ; \\
\Delta a_{1,1} &= 2q_4 \Delta q_4 - 2q_3 \Delta q_3 = -2q_4 \frac{q_3}{q_4} \Delta q_3 - 2q_3 \Delta q_3 = -4q_3 \Delta q_3; \\
\Delta a_{1,2} &= 2q_4 \Delta q_3 + 2q_3 \Delta q_4 = 2q_4 \Delta q_3 - 2q_3 \frac{q_3}{q_4} \Delta q_3 = \frac{2}{q_4} \left(1 - 2q_3^2\right) \Delta q_3.
\end{aligned} \tag{58}
$$

This means that the relationship (57) becomes:

$$
\Delta u_u = k_7 \left[ \left(1 - 2q_3^2\right) \Delta V_x + 2q_3 q_4 \Delta V_y + 4q_3 \left( V_y \frac{\left(1 - 2q_3^2\right)}{2q_3 q_4} - V_x \right) \Delta q_3 \right] + \Delta f_u. \tag{59}
$$

Having the quaternion-related functions linearized, we will look for the linear form of the guidance signal for linear coordinates in different evolutions.

For ascending/descending evolution, the linear coordinate signal in local frame (20) becomes:

$$
\begin{aligned}
\Delta u_x &= k_8 \Delta x_0 + k_9 \Delta V_x + \Delta f_x\ ;\ \Delta u_y = k_9 \Delta V_y + \Delta f_y\ ; \\
\Delta u_z &= k_8 z_0 + k_9 \Delta V_z + \Delta f_z,
\end{aligned} \tag{60}
$$

where $\Delta f_z$; $\Delta f_y$ is the input signal containing the desired linear coordinates.

Then, for $q_3 = \sqrt{2}/2$ corresponding base movement for $\theta = 90°$, we add axial velocity control (59):

$$
\Delta u_u = k_7 \Delta V_y + \Delta f_u\ . \tag{61}
$$

For horizontal evolution, the linear coordinate in the local frame (21) became:

$$
\begin{aligned}
\Delta u_x &= k_9 \Delta V_x + \Delta f_x\ ;\ \Delta u_y = k_8 \Delta y_0 + k_9 \Delta V_y + \Delta f_y; \\
\Delta u_z &= k_8 z_0 + k_9 \Delta V_z + \Delta f_z.
\end{aligned} \tag{62}
$$

Then, for $V_y = 0$ we add axial velocity control (58), (59):

$$\Delta u_u = k_7 \big[ (1 - 2q_3^2) \Delta V_x + 2q_3 q_4 \Delta V_y - 4q_3 V_x \Delta q_3 \big] + \Delta f_u. \tag{63}$$

If we neglected the time delay introduced by actuator Equations (25) and (34), we obtain a direct link between the linear form of the guidance signals and commands:

$$\Delta \delta_T = \Delta u_T; \ \Delta \delta_l = \Delta u_l; \Delta \delta_m = \Delta u_m; \ \Delta \delta_n = \Delta u_n. \tag{64}$$

In this case, starting from the scalar form of the guidance signals (18), we can write the linear form of the command components directly, separately, for the two motions.

The command for longitudinal motion in ascending/descending evolution becomes:

$$\begin{aligned} \Delta \delta_T &= -(k_9 + k_7) \Delta V_y \, ; \\ \Delta \delta_n &= -k_3 \Delta r - 2k_4 q_4^{-1} \Delta q_3 + k_8 \Delta x_0 + k_9 \Delta V_x. \end{aligned} \tag{65}$$

Denoting:

$$k_T^{Vy} = k_9 + k_7; \ k_n^{q3} = 2k_4 q_4^{-1}, \tag{66}$$

we obtain the regulator matrix indicated in Table 9.

**Table 9.** The regulator matrix for longitudinal motion in ascending/descending evolution *K*.

$$\begin{bmatrix} & k_T^{Vy} & & & \\ -k_9 & & k_3 & k_n^{q3} & -k_8 \end{bmatrix}$$

The command for longitudinal motion in horizontal evolution becomes:

$$\begin{aligned} \Delta \delta_T &= -(k_9 + k_7)(1 - 2q_3^2) \Delta V_x - 2(k_9 + k_7) q_3 q_4 \Delta V_y + 4(k_9 + k_7) V_x q_3 \Delta q_3 - \\ & \qquad 2k_8 q_3 q_4 \Delta y_0 \\ \Delta \delta_n &= -k_3 \Delta r - 2k_4 q_4^{-1} \Delta q_3 - k_{10}(1 - 2q_3^2) \Delta y_0 + 2k_9 q_3 q_4 \Delta V_x - \\ & \qquad k_9 (1 - 2q_3^2) \Delta V_y. \end{aligned} \tag{67}$$

Denoting:

$$\Delta k_T^{Vx} = (k_9 + k_7)(1 - 2q_3^2); \ k_T^{Vy} = 2(k_9 + k_7) q_3 q_4;$$

$$k_T^{q3} = -4(k_9 + k_7) V_x q_3 q_4; \ k_T^{y} = 2k_8 q_3 q_4; \ k_n^{Vx} = -2k_9 q_3 q_4;$$

$$k_n^{Vy} = k_9 (1 - 2q_3^2); k_n^{q3} = 2k_4 q_4^{-1}; \ k_n^{y} = k_8 (1 - 2q_3^2), \tag{68}$$

we obtain the regulator matrix shown in Table 10.

**Table 10.** The regulator matrix for longitudinal motion in horizontal evolution *K*.

$$\begin{bmatrix} k_T^{Vx} & k_T^{Vy} & & k_T^{q3} & k_T^{y} \\ k_n^{Vx} & k_n^{Vy} & k_3 & k_n^{q3} & k_n^{y} \end{bmatrix}$$

The commands for lateral motion in all evolutions are:

$$\begin{aligned} \Delta \delta_l &= -k_1 \Delta p - 2k_2 q_4 \Delta q_1 - 2k_2 q_3 \Delta q_2; \\ \Delta \delta_m &= -k_5 \Delta q - 2k_6 q_4 \Delta q_2 + 2k_6 q_3 \Delta q_1 + k_8 \Delta z_0 + k_9 \Delta V_z. \end{aligned} \tag{69}$$

Denoting:

$$k_l^{q1} = 2k_2 q_4; k_l^{q2} = 2k_2 q_3; \ k_m^{q1} = -2k_6 q_3; k_m^{q2} = 2k_6 q_4, \tag{70}$$

the regulator matrix for lateral motion is shown in Table 11.

**Table 11.** The regulator matrix for lateral motion $K$.

$$\begin{bmatrix} & k_1 & k_l^{q1} & k_l^{q2} & \\ -k_9 & k_5 & k_m^{q1} & k_m^{q2} & -k_8 \end{bmatrix}$$

To obtain the matrix terms, as shown in work [28], we can apply an LQR synthesis to an optimal regulator. The problem implies solving an algebraic Riccati equation.

Finally, the control coefficients introduced in control signals relations (18)–(21) can be obtained from the regulator's elements:

For longitudinal motion in ascending/descending evolution, $k_3, k_8, k_9$, are obtained directly, while the others are given by: $k_4 = q_4 k_n^{q3}/2$; $k_7 = k_T^{Vy} - k_9$.

For longitudinal motion in horizontal evolution, $k_3$ is obtained directly, while the others are given by: $k_4 = q_4 k_n^{q3}/2$; $k_8 = k_T^y (2q_3 q_4)^{-1}$; $k_9 = -k_n^{Vx}(2q_3 q_4)^{-1}$; $k_7 = k_T^{Vy}(2q_3 q_4)^{-1} - k_9$.

For lateral motion, $k_1, k_5, k_8, k_9$ are obtained directly, while the others are given by: $k_2 = k_l^{q1} q_4^{-1}/2$; $k_6 = k_m^{q2} q_4^{-1}/2$.

## 5. Model Parameters, Flight Scenarios, Results

### 5.1. Model Parameters

To build a performance evaluation methodology, including the quality of the guidance system, it is necessary to define a realistic LTV model. Since the data for such test vehicles is uncertain, we will refer to the information for classical launcher stages. For this purpose, we consider the Lox-Kerosene pair as fuel for the rocket engine, which is a common solution found in a series of classic (Soyuz) but also current launchers (Atlas V, Antares, Falcon 1, Falcon 9)

If we adopt an initial take-off mass, the ratio between the structural mass and the mass at the start is defined by the structural coefficient [29]:

$$\varepsilon = \frac{m_s}{m_p + m_s}, \tag{71}$$

where $m_s$ is the structural mass and $m_p$ is the propellant mass.

As shown in [29], this coefficient measures the vehicle designer's skill and evolved over time for different launchers. Thus, if for the stages of the Soyuz launcher, the structural coefficient is around 9% for Falcon 1, it becomes 6%, and for Falcon 9, it becomes 3%, which shows the technological evolution of the launchers. In order to have a realistic model for LTV, we will consider a structural coefficient of 30%, which is much oversized and will allow the introduction of additional control and testing elements, specific to a test vehicle, into the structural mass.

Based on the previous statements, we will consider a hypothetical LTV with an initial mass of 100 kg, from which propellant is 70 kg, equipped with a liquid rocket engine with a specific impulse $I_{sp} = 230 [s]$, specific for the regular liquid engine (Kerosene + LOX), with maximum thrust $T_0 = 1200 [N]$ and reactive force of one RCS element $R = 5 [N]$.

The vehicle has the shape and the dimension presented in Figure 6.

From a dimensional point of view, we can check if the volume of the imposed fuel is compatible with the dimensions indicated in Figure 6.

For this, we start from the density of LOX of 1141 kg/m$^3$ and Kerosene of 825 kg/m$^3$. Considering the LOX/Kerosene mixture ratio = 2.72, we obtain an average propellant density of 1056 kg/m$^3$, which for a mass of 70 kg leads to a hypothetical spherical tank with a radius of 0.25 m, a size compatible with the geometric dimensions of the vehicle indicated in Figure 6.

For the considered configuration, the reference length is $l = 1$ m, the reference surface $S = 0.283$ m$^2$ and the position of aerodynamic focus is $x_F = 0.4$ m.

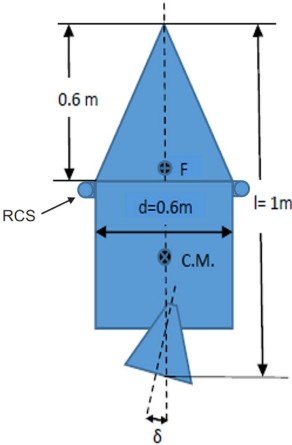

**Figure 6.** LTV sketch: C.M.—Center of Mass; F—Focus; RCS—reaction control system.

(a) *Mechanical data*

For mechanical data, we will consider two cases. The first will be the LTV at the start, with full fuel, and the second at the end of evolution, without fuel. Because the fuel consumption defined through Equation (7) depends on flight conditions, the intermediate values of the other mechanical data will be obtained through interpolation as a function of LTV mass. The mechanical characteristics of LTV are shown in Table 12.

**Table 12.** Mechanical characteristics.

| Phase | Mass m [kg] | Center of Mass $x_{cm}$ [m] | Roll Inertial Moment A [kgm$^2$] | Yaw Inertial Moment B kgm$^2$ | Pitch Inertial Moment C kgm$^2$ |
|---|---|---|---|---|---|
| Initial | 100 | 0.666 | 10 | 15 | 15 |
| Final | 30 | 0.456 | 3 | 7 | 7 |

(b) *Time constants and controller gains*

The time constants for actuators are: $\tau_T = 0.5$ s; $\tau_n = 0.1$ s; $\tau_l = 0.1$ s; $\tau_m = 0.1$ s and the gains used in control signals are: $k_1 = 3.9$; $k_2 = 2.5$; $k_3 = 2.0$; $k_4 = 5.5$; $k_5 = 2.2$; $k_6 = 6.9$; $k_7 = 0.32$; $k_8 = 0.7$; $k_9 = 1$. Model parameters defined will be used for the development of subsequent applications. For the evaluation of the guidance precision, an uncertainty of these parameters will be considered, which will lead to a dispersion of the evaluated trajectories, including the impact point.

*5.2. Flight Scenarios*

Although in some papers [12,13], the optimal trajectories for reaching a desired final position are analyzed, in the initial phase of LTV testing, it is necessary to define some simple trajectories to verify the vehicle's controllability as its field of use by defining the flight envelope.

For this purpose, we will further define two flight scenarios, one vertical ascent/descent type, and the second having a horizontal evolution interspersed between the vertical evolutions of ascent/descent.

In the first scenario, LTV starts at an altitude $y_0 = 0.1$ [m], with an initial velocity $V_{y0} = 0.1$ [ms$^{-1}$] and initial pitch angle: $\theta_0 = 90°$. Next, the ascending phase follows, which lasts until LTV achieves the desired altitude $y_d$, evolving at an imposed pitch angle $\theta_{d1} = 90°$ and imposed ascensional velocity $V_{yd1} = V$. After ascending phase, the descending phase follows with imposed pitch angle $\theta_{d2} = 90°$, evolution being vertical with descending constant velocity $V_{yd2} = -V$, which lasts until it reaches the breaking

altitude $y_f$. After $y_f$ braking occurs, and the speed decreases until $V_{y4} = 1 \left[\text{ms}^{-1}\right]$ ensuring a smooth landing for the LTV.

In the second scenario, LTV starts at an altitude $y_0 = 0.1 \, [\text{m}]$, with an initial velocity $V_{y0} = 0.1 \left[\text{ms}^{-1}\right]$ and initial pitch angle: $\theta_0 = 90°$. Next, the ascending phase follows, which lasts until LTV achieves the desired altitude $y_d$ evolving at an imposed pitch angle $\theta_{d1} = 90°$ and imposed ascensional velocity $V_{yd1} = V$. After the ascending phase, the horizontal phase follows at an imposed altitude $y_d$, with $V_{yd2} = 0$, until LTV achieves the desired abscissa $x_d$, with imposed velocity $V_{xd} = V$. After the horizontal phase, the descending phase follows with imposed pitch angle $\theta_{d2} = 90°$, evolution being vertical with descending constant velocity $V_{yd3} = -V$ which lasts until it reaches the breaking altitude $y_f$. After $y_f$ braking occurs, and the speed decreases until $V_{y4} = 1 \left[\text{ms}^{-1}\right]$ ensuring a smooth landing for the LTV.

To synthesize the flight evaluation, we chose to have the same module of velocity $V$ in the entire flight evolution, except for the breaking phase when the value is $1 \left[\text{ms}^{-1}\right]$. The value of flight velocity $V$ will be particularized for the test case.

*5.3. Results*

(a)   *First test case*

In order to exemplify the first scenario, we will consider an evolution with $V = 10 \left[\text{ms}^{-1}\right]$, with the desired altitude $y_d = 400 \, [\text{m}]$. Figure 7 presents the vertical trajectory obtained by LTV for this first test case. The trajectory contains two flight phases: ascending phase and descending phase.

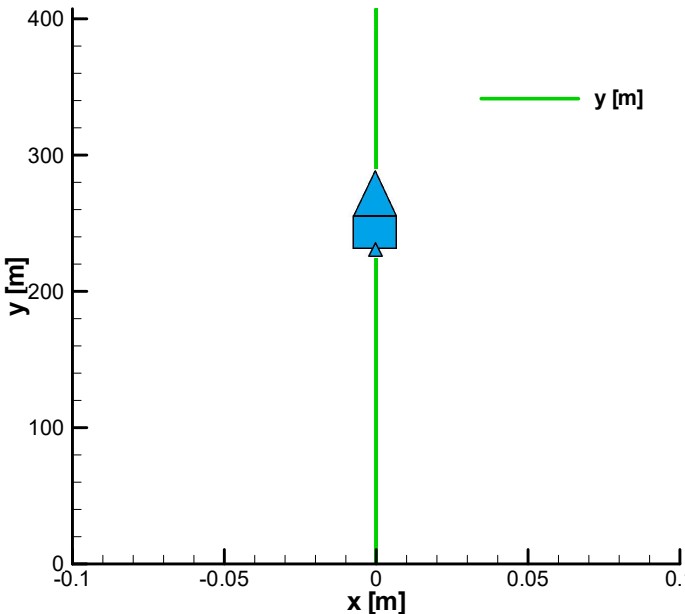

**Figure 7.** Vertical ascending—descending trajectory.

Figure 8 shows the equilibrium solutions of Equations (40) and (41) (*q3_e, dt_e, dnt_e*) compared to the solutions obtained through differential equations integration for the 6DOF model in the first scenario (*q3, dt, dnt*).

Figure 9 shows eigenvalues of the stability matrix *A* for longitudinal motion in ascending/descending evolution corresponding to Table 3. We can observe a pair of complex eigenvalues with real part positive due to static instability (position aerodynamics focus in front of the mass center). The variation of the eigenvalues corresponds with the variation of the flight parameters in the first scenario.

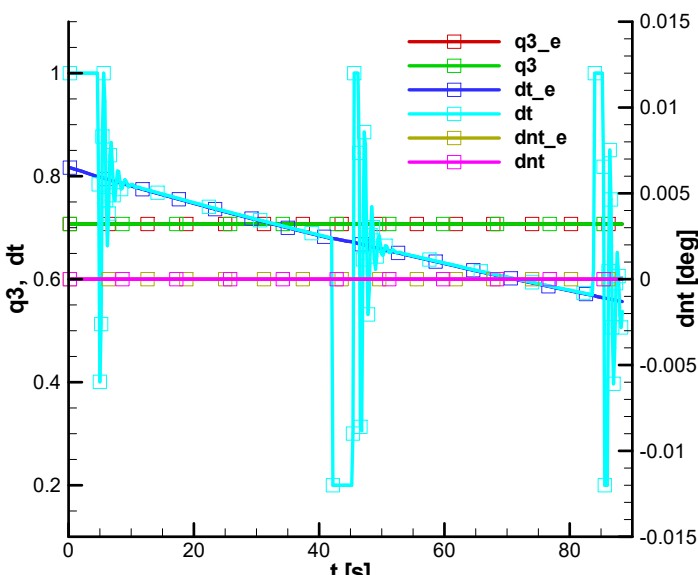

**Figure 8.** The equilibrium commands and *q3* component for evolution in the first scenario.

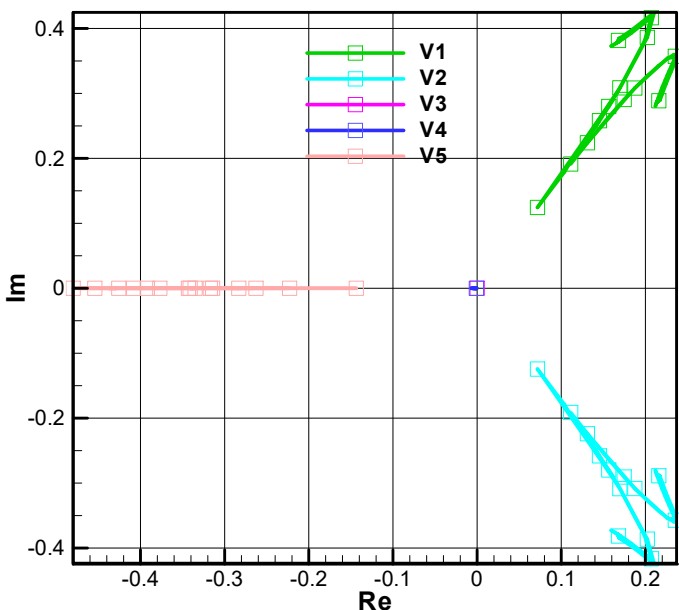

**Figure 9.** Eigenvalues of the stability matrix A for longitudinal motion in ascending/descending evolution.

Figure 10 shows the eigenvalues of the regulated matrix $A - BK$ for longitudinal motion in ascending/descending evolution. Considering the variation of the eigenvalues for the open loop system in Figure 9, one can observe that the real part of all eigenvalues for the closed-loop system is negative, resulting in stability for the first scenario evolution.

Figure 11 shows the eigenvalues of the stability matrix $A$ for the uncontrolled lateral motion corresponding to Table 7. We can observe a pair of complex eigenvalues with real part positive due to static instability (position aerodynamics focus in front of the mass center). The variation of the eigenvalues corresponds to the variation of the flight parameters in the first scenario. Due to the symmetry of the configuration and evolution, the shown eigenvalues of the stability matrix for the lateral motion correspond to the eigenvalues of the stability matrix $A$ for longitudinal motion presented in Figure 9. The difference consists in the number of eigenvalues. For lateral motion, we have six values; for longitudinal, we only have five values.

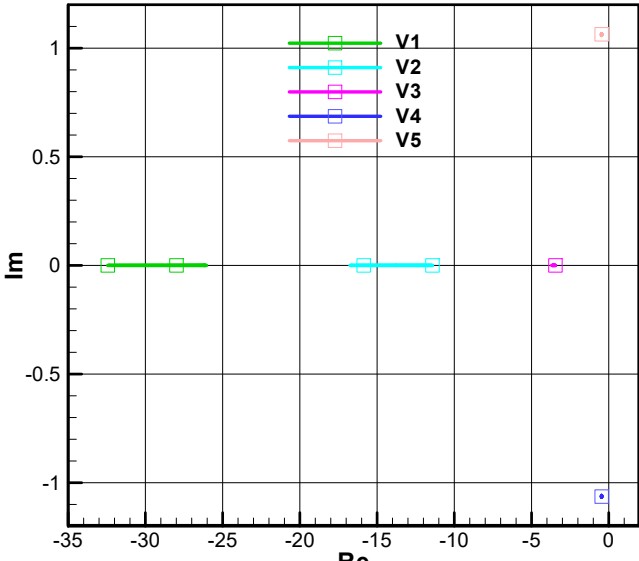

**Figure 10.** Eigenvalues of the matrix A − BK for longitudinal motion in ascending/descending evolution.

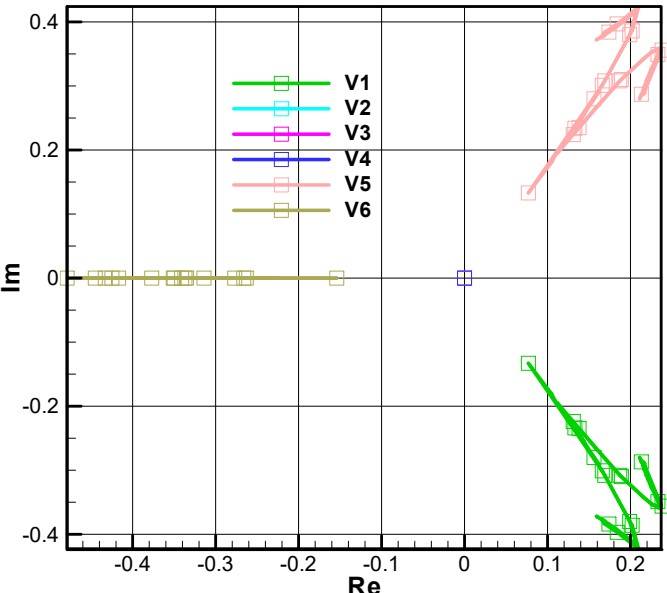

**Figure 11.** Eigenvalues of the stability matrix $A$ in lateral motion.

Figure 12 shows eigenvalues of the regulated matrix $A − BK$ for lateral motion in all evolutions. One can observe that the real part of all eigenvalues has real negative parts, implying the stability of this evolution. Due to the symmetry of the configuration and evolution, the shown eigenvalues of the regulated matrix for lateral motion correspond to the eigenvalues of the regulated matrix $A − BK$ for longitudinal motion presented in Figure 10. The difference consists in the number of the eigenvalues. For lateral motion, there are six values, while for longitudinal only five.

From Figure 13, we can observe that vertical velocity $Vy$ increases during ascending phase to 10 m/s and becomes negative ($-10$ m/s) during the descending phase and finally has the value of $-1$ m/s during the breaking phase. Moreover, we can observe the vertical coordinate $y$ that increases from 0.1 m to 400 m and then decreases to zero.

From Figure 14, we observe that the thrust throttling command $dT$ decreases during ascending and descending phases, following the corresponding decrease in LTV mass. The command reaches peak values during transition phases.

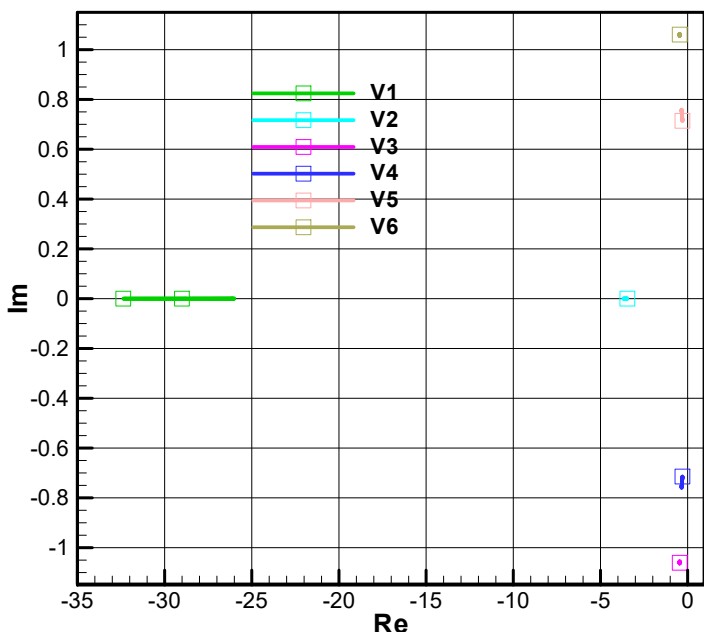

**Figure 12.** Eigenvalues of the matrix A − BK for lateral motion.

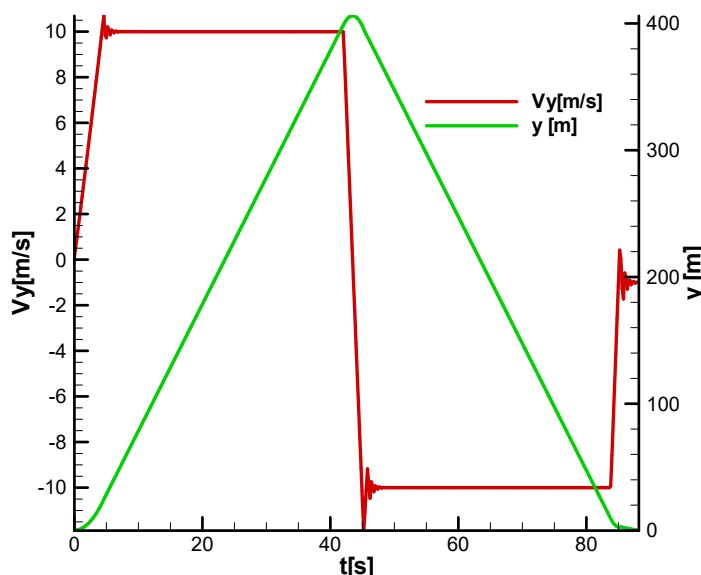

**Figure 13.** Vertical velocity and altitude.

In terms of lateral and rolling movement, the values of states and commands are null during all flight phases.

The lateral and roll channels have no state or command variations (their derivatives remain null) during the first flight scenario.

(b)  *Second test case*

In order to exemplify the second scenario, we will consider an evolution with $V = 7$ [m/s], at altitude $y_d = 300$ m with the imposed distance of the horizontal flight $x_d = 1000$ m.

Figure 15 presents the vertical trajectory obtained by LTV. We can observe three flight phases: ascending, horizontal, and descending. We can also observe that the flight attitude for LTV in all three phases is with the pitch angle close to 90 degrees ($q_3 = 0.707$) corresponding to the basic movement shown in Figure 16.

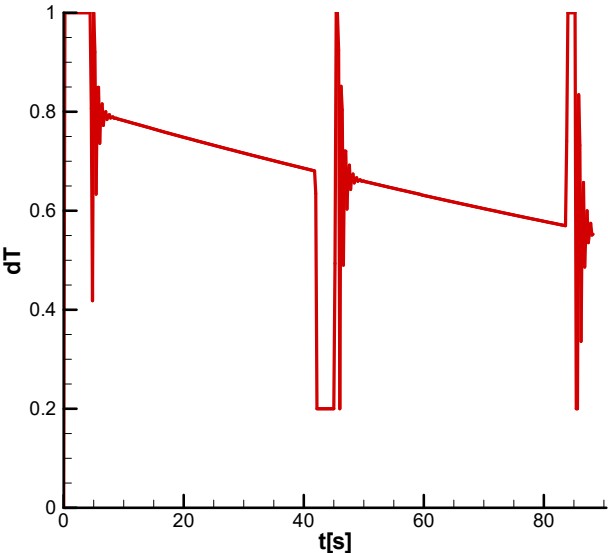

**Figure 14.** Diagram of the thrust throttling command *dT*.

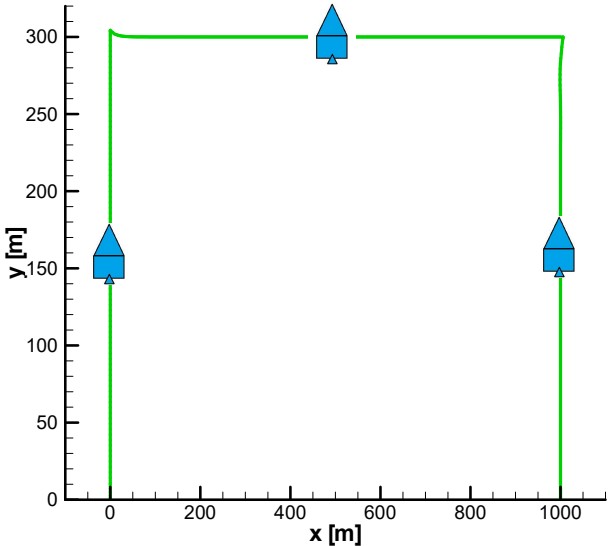

**Figure 15.** Vertical trajectory with horizontal flight.

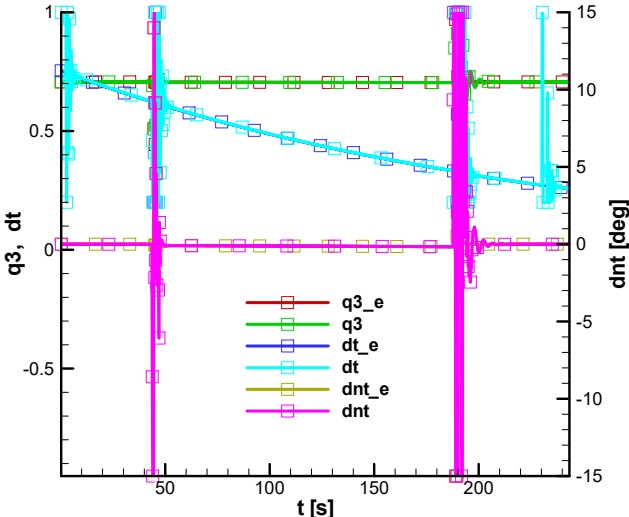

**Figure 16.** The equilibrium commands and q3 component for evolution in the second scenario.

Figure 16 shows the equilibrium solutions of Equations (40) and (41) (*q3_e, dt_e, dnt_e*) compared to the solution obtained through the integration of differential equations for the 6 DOF model in the second scenario (*q3, dt, dnt*).

Figure 17 shows eigenvalues of the stability matrix *A* for longitudinal motion in horizontal evolution corresponding to Table 5. We can observe a pair of complex eigenvalues with real part positive due to static instability (position of the aerodynamics focus in front of the mass center). The variation of the eigenvalues corresponds to the variation of the flight parameters in horizontal evolution in the second scenario.

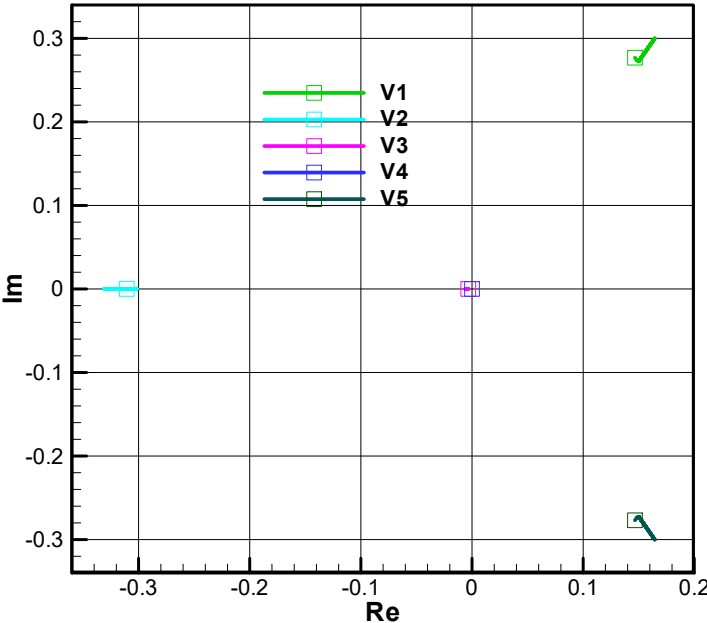

**Figure 17.** Eigenvalues of the stability matrix *A* for longitudinal motion in horizontal evolution.

Figure 18 shows eigenvalues of the regulated matrix *A* − *BK* for longitudinal motion during horizontal evolution. Taking into consideration the variation of the eigenvalues for the open loop system in Figure 17, one can observe that the real part of all eigenvalues for the closed-loop system is negative, resulting in stability for the entire evolution for the second scenario.

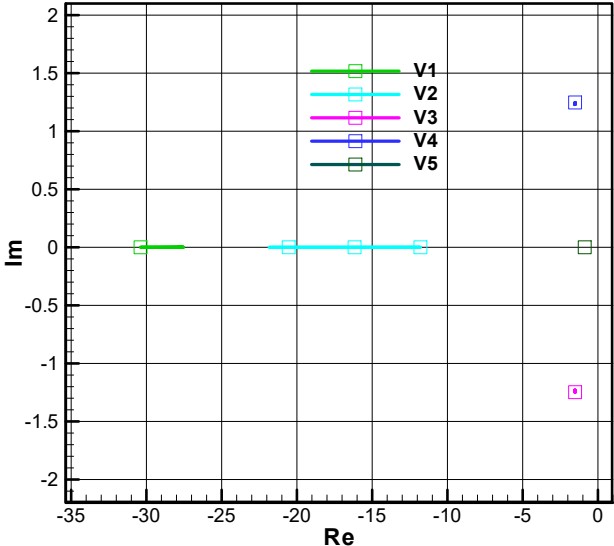

**Figure 18.** Eigenvalues of the matrix *A* − *BK* for longitudinal motion in horizontal evolution.

Figure 19 shows vertical velocity and vertical coordinate (altitude). We can observe that vertical velocity Vy increases during ascending phase until 7 m/s and is null in horizontal evolution, then becomes negative $\left(V_y = -7\,\text{m/s}\right)$ during the descending phase and finally has the value $V_y = -1$ m/s during the breaking phase. Moreover, we can observe the vertical coordinate y that increases from 0.1 [m] to 300 [m] and decreases to zero.

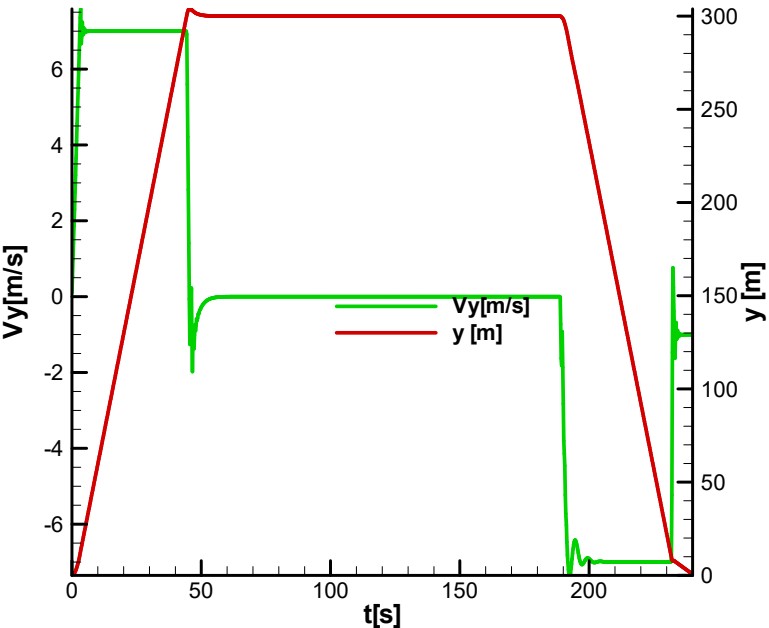

**Figure 19.** Vertical velocity and altitude.

From Figure 20, we can observe that horizontal velocity $Vx$, which is zero during the ascending phase, has a value of 7 [m/s] in horizontal evolution and then becomes zero during the descending and breaking phases. Moreover, we can observe the abscissa $x$ that increases from 0 m until 1000 m.

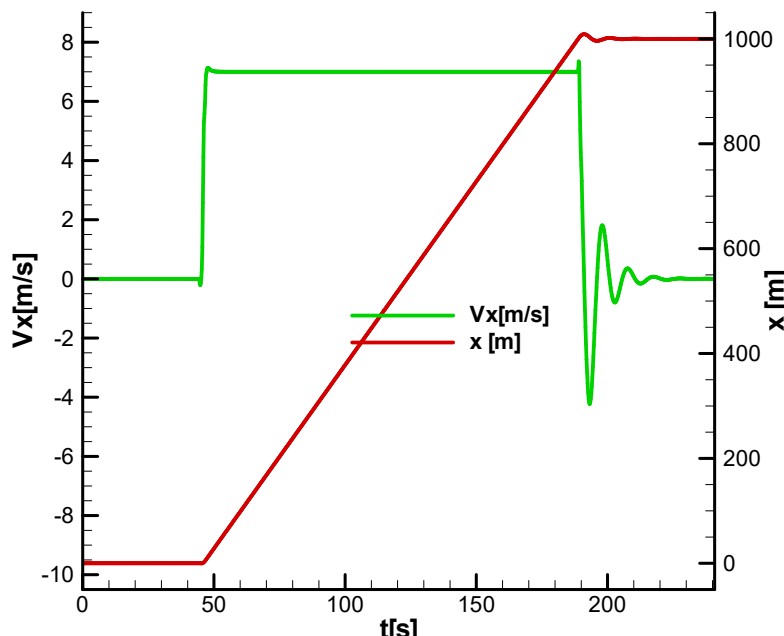

**Figure 20.** Horizontal velocity and horizontal coordinate.

Figure 21 shows the velocity components in the local frame. We can observe that vertical velocity $V_y$ increases during the ascending phase, null in the horizontal evolution and becomes negative during the descending phase. The horizontal velocity $Vx$ is null during the ascending phase, becomes constant during the horizontal phase, and null during the descending phase.

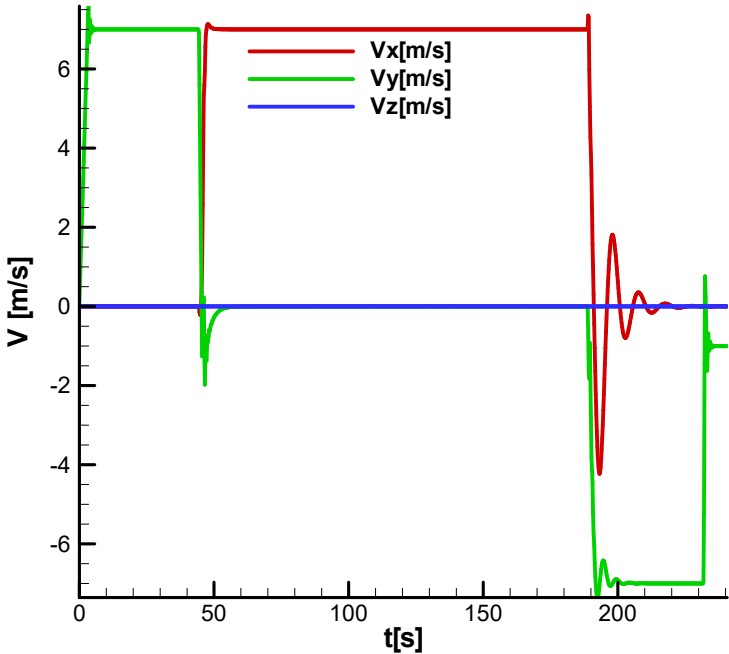

**Figure 21.** Velocity components along the local frame.

Even though a quaternion approach is used in the 6DOF model, we prefer to present the results using Euler angles for a better understanding. Figure 22 shows the desired pitch angle (*ted*) and the achieved pitch angle (*te*). We can observe that in ascending phase, the pitch angle has the value $\theta_{d1} = 90°$, during horizontal evolution, follows the value of the equilibrium pitch angle, which becomes smaller while the mass decreases, and in descending phase, it takes the value $\theta_{d2} = 90°$. Figure 23 shows a detail of Figure 22.

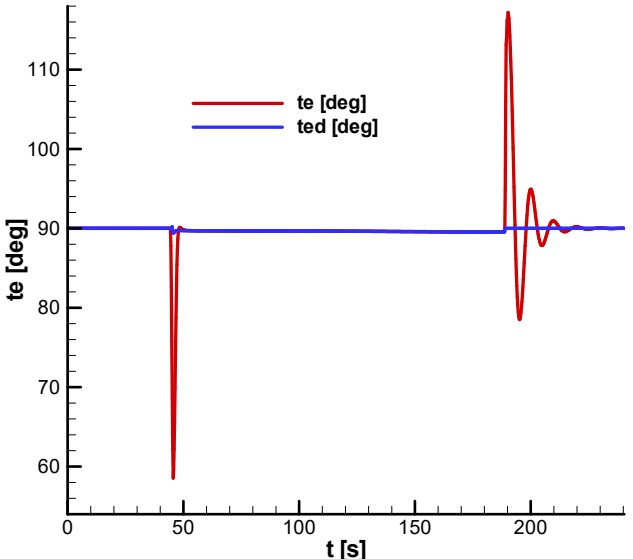

**Figure 22.** Diagram of the pitch angle: *ted*—desired pitch angle; *te*—achieved pitch angle.

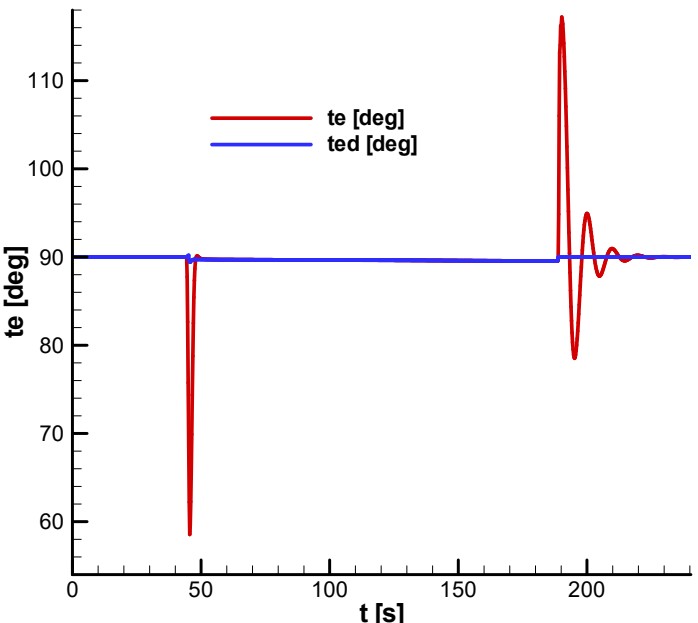

**Figure 23.** Diagram of the pitch angle: *ted*—desired pitch angle; *te*—achieved pitch angle (detail).

Figure 24 shows the longitudinal commands. We observe that thrust throttling command dT has maximum value during the ascending phase, after which it decreases continuously during horizontal and descending evolutions. Pitch deflection command dn is null during the ascending phase, becomes negative during horizontal evolution, and is null during descending evolution.

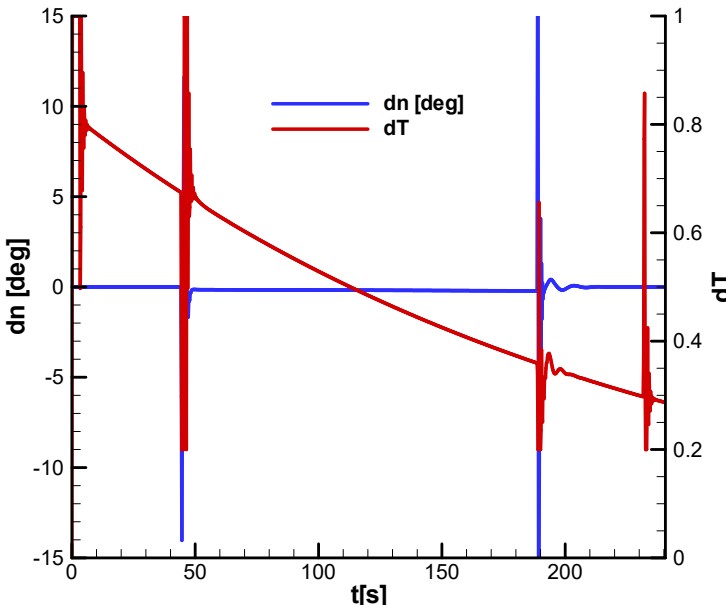

**Figure 24.** Diagram of longitudinal commands *dT*—thrust throttling command; *dn*—pitch deflection command.

Figure 25 shows the incidence angle *alfa* and the pitch angle *te*. We can observe that incidence reaches the value of the pitch angle after the ascending phase, then maintains a value equal to the value of the pitch angle during horizontal evolution, and during the descending phase, it gets close to 180 degrees due to vertical descending evolution. The incidence angles and related pitch angle values prove that vehicle maintains a vertical attitude with the tip up in all flight phases.

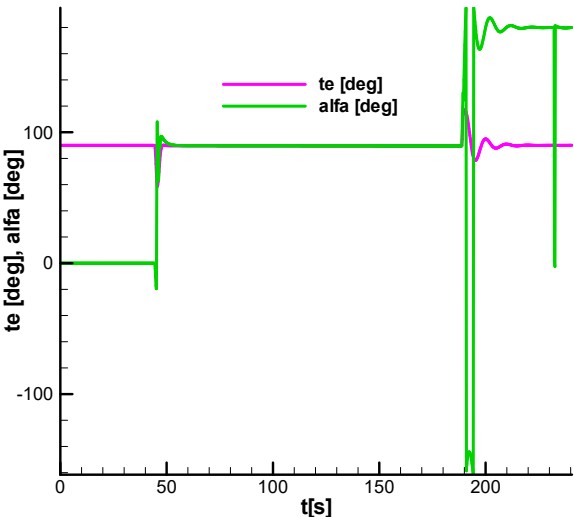

**Figure 25.** Diagram of the pitch angle (*te*) and the incidence angle (*alfa*).

In terms of lateral and roll motion, the values of states and commands are null during all the flight phases.

During the second flight scenario, the lateral and roll channels have no state or command variations (their derivatives remain null).

## 6. Performances

### 6.1. Flight Envelopes

We will evaluate the vehicle's performance based on the previously presented scenarios. According to [19], the performance of the guided vehicle means the flight envelope and the guidance precision. Next, we will evaluate the flight envelope using the previous described scenarios by determining maximum altitude and distance. Then, considering the uncertainty of the model parameters and sensor noise, we will evaluate the trajectory dispersions.

(a)    *Maximum altitude*

Considering the first scenario, the vertical evolution, and using different ascending/descending velocities, we can obtain the maximum altitude of the vehicle as a velocity function, presented in Figure 26.

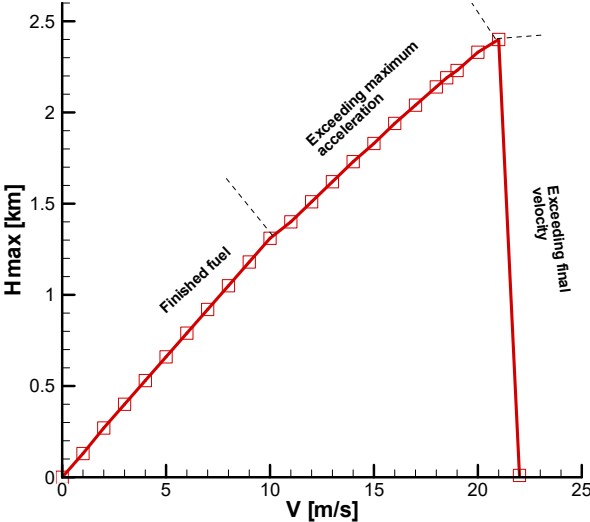

**Figure 26.** Maximum altitude for vertical evolution—first scenario.

From Figure 26, we can observe that maximum altitude increases with velocity until it reaches a peak after that decreases. The maximum altitude obtained by the vehicle was $Hmax = 2.4$ km, corresponding to $V = 21$ [m/s]. Moreover, from Figure 26, we can observe three restrictions that limit the flight envelope: fuel consumption; exceeding maximum acceleration during maneuver $\left(a_{max} = 30\ [\text{m/s}^2]\right)$; exceeding final velocity $\left(V_{final\ max} = 2\ [\text{m/s}]\right)$.

(b)　*Maximum distance*

Considering the second scenario, with horizontal evolution, and using different velocities and different altitudes for the horizontal flight, we can obtain the maximum distance of the vehicle as a velocity and altitude function is presented in Figure 27.

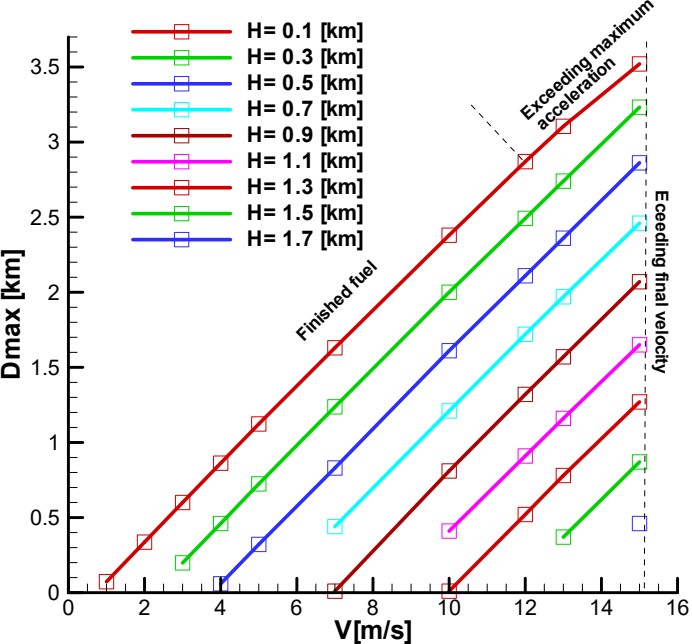

**Figure 27.** Maximum distance for horizontal evolution—second scenario.

Figure 27 shows that maximum distance increases with velocity and decreases with the horizontal flight altitude. The maximum distance obtained by vehicle was $Dmax = 3.52$ km, corresponding to $V = 15$ [m/s] and an altitude of horizontal flight $H = 0.1$ [km]. Additionally, from Figure 27, we can observe a restriction of the flight envelope due to the exceedance of the final velocity $\left(V_{final\ max} = 2\ [\text{m/s}]\right)$, fuel consumption and exceedance of maximum acceleration $\left(a_{max} = 30\ [\text{m/s}^2]\right)$, the same restriction as in the case of maximum altitude.

### 6.2. Wind Influence

For wind influence, we consider the second scenario, uniform wind and turbulence, as described in [17]. First, we consider lateral wind, with the velocity $Wz = 1$ m/s, the influence of lateral parameters being presented in the next figure (Figure 28).

Further, we consider a layer of turbulence between the altitude of 200–250 m. This layer of turbulence influences the motion in the ascending and descending phases, as seen in Figure 29.

The results prove that the vehicle is stable when considering the wind influence, lateral deviation in $z$ coordinate, and $Vz$ velocity in both cases being insignificant.

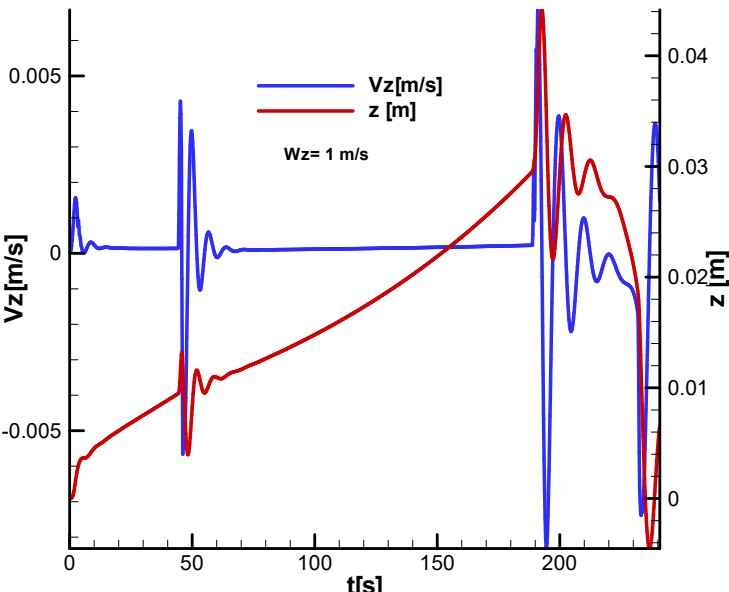

**Figure 28.** Influence of lateral wind on the lateral coordinate (*z*) and corresponding velocity (*Vz*).

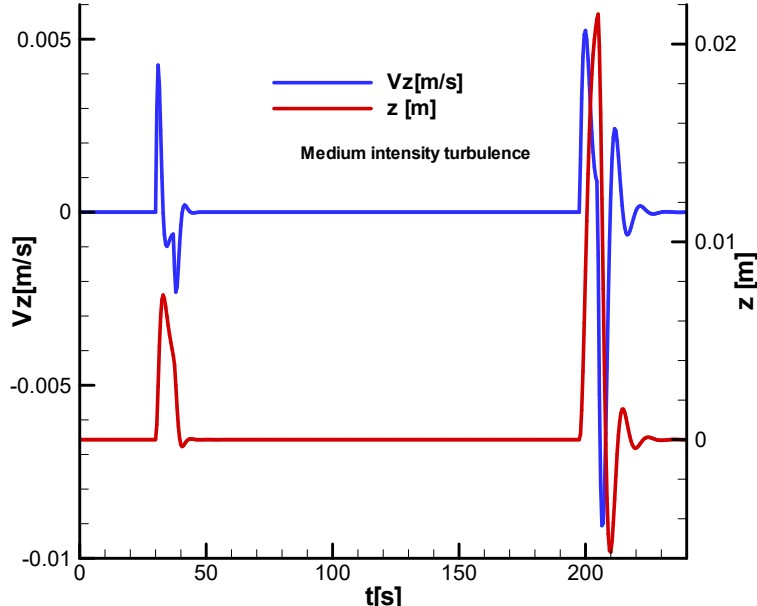

**Figure 29.** Influence of turbulence on the lateral coordinate (*z*) and lateral velocity (*Vz*).

### 6.3. Flight Parameters Dispersion

To evaluate dispersion of flight parameters we consider the dispersion with normal distribution of some uncertain model parameter, defined in Section 2.5 and Section 5.1, as a percentage of the nominal value, as follows:

Dispersions of aerodynamic drag coefficient and position of the aerodynamic focus related reference length:

$$D_A = E_A (C_{ds})^2 ; D_X = E_X \left( \tilde{x}_f^2 \right)^2 . \tag{72}$$

Dispersion of mass and inertial moments:

$$D_m = E_m m^2 ; D_I = E_I C^2 . \tag{73}$$

For LTV, the following values for standard deviations were considered:

$$\sqrt{E_A} = 5\%; \sqrt{E_x} = 5\%; \sqrt{E_m} = 2\%; \sqrt{E_I} = 2\%. \tag{74}$$

The uncertainty of the model parameters described previously are considered input data and do not change over time during the flight.

Supplementary an additive noise affecting sensors measurement having uniform distribution was considered: sensors for angular attitude $\pm 1/60$ *deg*; sensors for linear position $\pm 0.3$ m.

The additive noise of the sensor will be variable in time but maintain uniform distribution.

To obtain the trajectory dispersion, we ran 500 tests for each flight case presented previously, the values for uncertain sizes and noise for the sensor being obtained by random number generators [30].

Figure 30 shows a trajectory beam in a vertical plane corresponding to the first case.

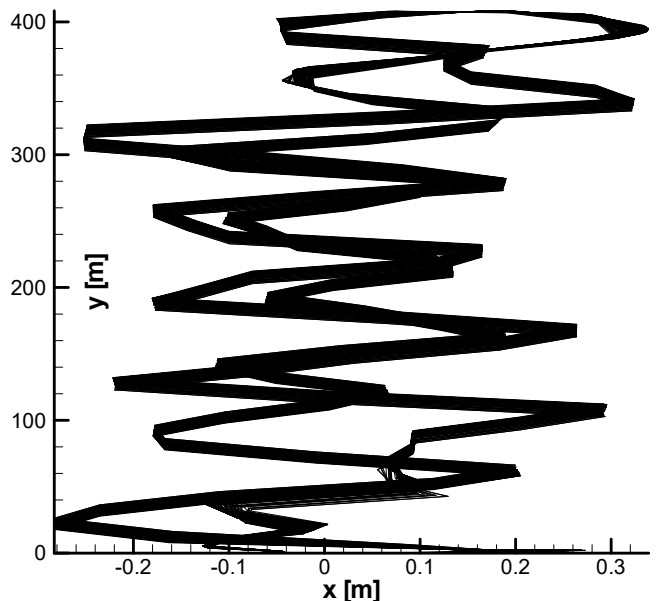

**Figure 30.** Trajectory beam in lateral view, case 1.

Figure 31 shows a trajectory beam in a horizontal plane corresponding to the first case.

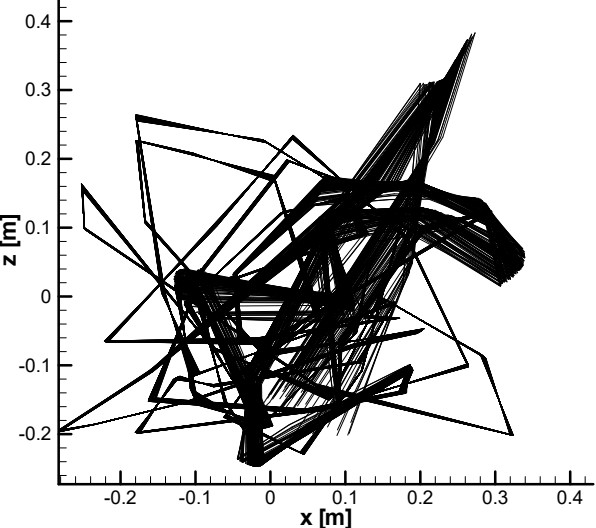

**Figure 31.** Trajectory beam in top view, case 1.

Synthesizing the results for landing positions in Table 13, we can observe that the final coordinate average does not exceed the desired values ($X = 0$, $Z = 0$) with more than a few centimeters, and standard deviations are also small values. Regarding final velocity, this is very close to the desired values ($V = 1$ m/s), and the standard deviation has a small value (12 cm/s).

**Table 13.** Statistic elements of the landing position in case 1.

| Phase | T [s] | V [m/s] | X [m] | Z [m] |
| --- | --- | --- | --- | --- |
| Average | 88.28 | 1.053 | −0.085 | −0.203 |
| Standard deviation | 0.416 | 0.128 | 0.122 | 0.072 |

Figure 32 shows a trajectory beam in a vertical plane corresponding to the second case.

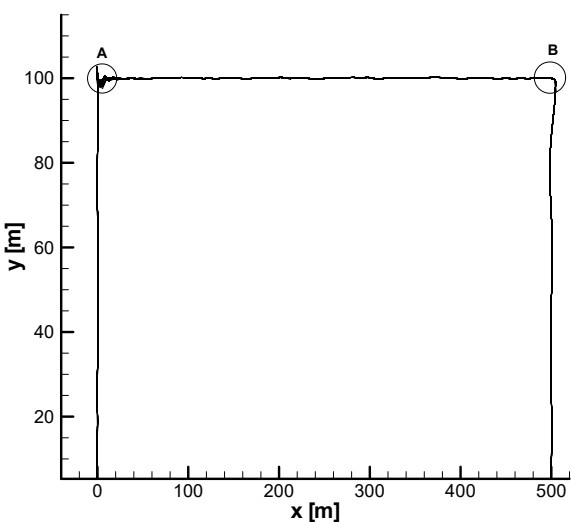

**Figure 32.** Trajectory beam in lateral view, case 2.

In Figures 33 and 34, we present the trajectory details (A,B) in the vertical plane to highlight the trajectory dispersion in transition phases between vertical and horizontal evolution.

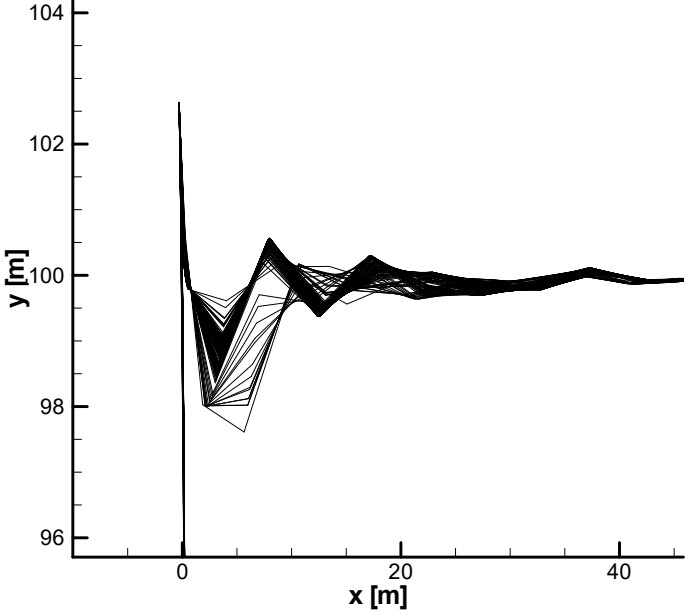

**Figure 33.** Trajectory beam in lateral view, case 2—detail A.

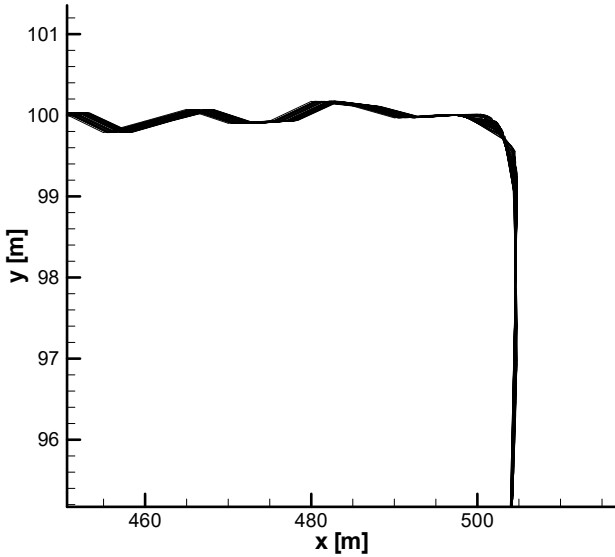

**Figure 34.** Trajectory beam in lateral view, case 2—detail B.

Figure 35 shows a trajectory beam in a horizontal plane corresponding to the second case.

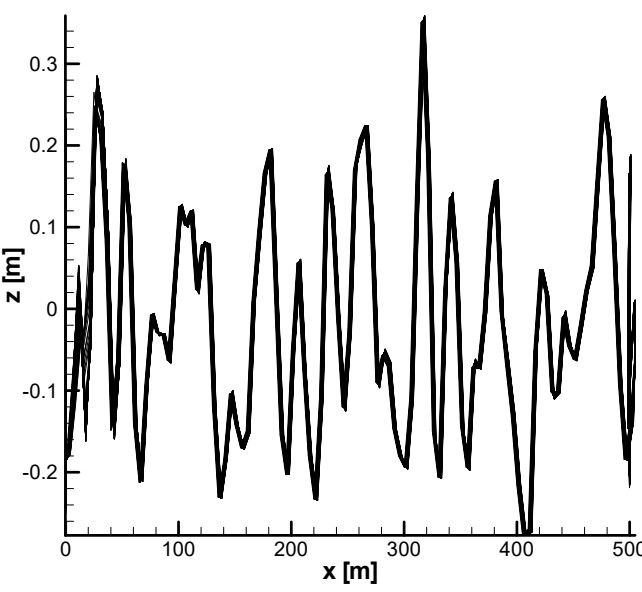

**Figure 35.** Trajectory beam in top view, case 2.

Synthesizing the results for landing positions (Tables 13 and 14), we can observe that the average of the final coordinate does not exceed the desired values ($X = 500$ m, $Z = 0$ m) with more than a few centimeters, and standard deviations are also small values. Regarding final velocity, this is very close to the desired value ($V = 1$ m/s), and the standard deviation has a very small value (0.8 cm/s).

**Table 14.** Statistic elements of the landing position in case 2.

| Phase | T [s] | V [m/s] | X [m] | Z [m] |
|---|---|---|---|---|
| Average | 151.47 | 1.01 | 499.97 | −0.02 |
| Standard deviation | 1.248 | 0.008 | 0.17 | 0.021 |

## 7. Conclusions

In order to build the equation of motion for the testing vehicle, we use two frames. The first one is the Local Frame/Start Frame, which allows us to write the translational equations; the second is the Body Frame, which allows us to write rotational dynamic equations. In order to correlate the model and the result of the testing vehicle with the model and results of the launcher, we use the Start Frame with the y-axis up and quaternion Hamilton. Further, we write the relations that describe the guidance command, which allows for building a 6DOF guided model. To complete the definition, aerodynamics, and thrust terms were introduced. Starting from the 6DOF nonlinear model, we obtained the basic movement, the linear form of the motion equation, and the stability and command matrices. From the analysis of the basic movement, it was found that for the technical solution adopted, the flight attitude is always vertical. Figure 15 shows that it leads to horizontal evolution with a high incidence angle, close to 90 degrees. Using the model, two flight scenarios were evaluated. The first one contains only ascending and descending evolution, and the second contains three phases: ascending, horizontal, and descending. For each scenario, a flight case was defined and analyzed, and the flight envelope of the testing vehicle was defined. Supplementary, the influence of the uncertainty of the model parameters and sensor noise on trajectory dispersion was evaluated. Moreover, the influence of uniform wind and turbulence on lateral trajectory deviation was analyzed. The model developed must be improved by using experimental measurements of the dynamic regime's aerodynamic terms and thrust characteristics. After completing ground measurements, the 6DOF model can be used for flight experiments design. In order to achieve the control system, it is necessary to use the inertial measurement unit (IMU) combined with GPS measurements to obtain information on the position, as well as the attitude and angular rate of the vehicle. The developed model for LTV, which is similar to the launcher model, can be used to validate the reusable launcher GNC in ascending/descending and horizontal flight phases.

In summary, the novelty element of the paper, and at the same time, the contributions in the field are:

- Building a complex computational model dedicated to the autonomous flight of the LTV;
- Defining the basic movement for the LTV and checking its concordance with the numerical solution obtained for autonomous flight Figures 8 and 16;
- Obtaining the linear form of the equations of motion in the local frame with the exclusive use of the quaternion and the cross-check between the analytical and the numerical solution;
- Obtaining the linear form of the guidance relations and using them to build the regulator matrix in the required form;
- Methodology for evaluation of LTV performances using two flight scenarios.

The proposed model can be improved especially in the guidance part regarding the transition from one flight phase to another, where the state variables are observed in strong oscillations.

**Author Contributions:** Conceptualization, T.-V.C.; methodology, T.-V.C.; software, T.-V.C.; validation, T.-V.C.; formal analysis, T.-V.C., V.P. and C.E.; investigation, T.-V.C.; resources, V.P.; data curation, T.-V.C., V.P. and C.E.; writing—original draft preparation, T.-V.C.; writing—review and editing, V.P. and C.E.; visualization, V.P. and C.E.; supervision, T.-V.C.; project administration, V.P. All authors have read and agreed to the published version of the manuscript.

**Funding:** This work has been funding by University POLITEHNICA of Bucharest through the PubArt program.

**Institutional Review Board Statement:** Not applicable.

**Informed Consent Statement:** Not applicable.



**Data Availability Statement:** Data is contained within the article.

**Conflicts of Interest:** The authors declare no conflict of interest.

**Nomenclature**

| | |
|---|---|
| $q_1, q_2, q_3, q_4;$ | Quaternion components; |
| $\rho$ | Air density; |
| $m$ | Mass; |
| $A, B, C$ | Inertial moments; |
| $\boldsymbol{G}$ | Weight |
| $\boldsymbol{g}$ | Gravity acceleration; |
| $\boldsymbol{F}$ | Aerodynamic force; |
| $\boldsymbol{H}$ | Aerodynamic momentum; |
| $X^A; Y^A; Z^A$ | Aerodynamic force components in body frame; |
| $L^A; M^A; N^A;$ | Aerodynamic momentum components in body frame; |
| $\boldsymbol{T}$ | Thrust force; |
| $\boldsymbol{U}$ | Thrust momentum; |
| $X^T; Y^T; Z^T$ | Thrust force components in body frame; |
| $L^T; M^T; N^T$ | Thrust momentum components in body frame |

**Appendix A. Coupled form of the Linear Equations of Motion**

The objective of the appendix is to obtain the coupled linear form of the equations of motion. The basic movement considered is a translation in a vertical plane. This appendix uses the same nomenclature as the main paper.

*A.1. Translation Movement*

Starting from the nonlinear equations for translation in local frame:
The dynamic equations:

$$
\begin{bmatrix} \dot{V}_x \\ \dot{V}_y \\ \dot{V}_z \end{bmatrix} = m^{-1} \boldsymbol{B}_I \left( \begin{bmatrix} X^A \\ Y^A \\ Z^A \end{bmatrix} + \begin{bmatrix} X^T \\ Y^T \\ Z^T \end{bmatrix} \right) + \boldsymbol{g}_0. \tag{A1}
$$

The kinematic equations:

$$
\begin{bmatrix} \dot{x}_0 & \dot{y}_0 & \dot{z}_0 \end{bmatrix}^T = \begin{bmatrix} V_x & V_y & V_z \end{bmatrix}^T. \tag{A2}
$$

The linear form of the translation equations becomes:
The dynamic equations:

$$
\begin{bmatrix} \Delta \dot{V}_x \\ \Delta \dot{V}_y \\ \Delta \dot{V}_z \end{bmatrix} = m^{-1} \left\{ \Delta \left( \boldsymbol{B}_I \begin{bmatrix} X^A \\ Y^A \\ Z^A \end{bmatrix} \right) + \Delta \left( \boldsymbol{B}_I \begin{bmatrix} X^T \\ Y^T \\ Z^T \end{bmatrix} \right) \right\}. \tag{A3}
$$

The kinematic equations:

$$
\begin{bmatrix} \dot{\Delta x}_0 & \dot{\Delta y}_0 & \dot{\Delta z}_0 \end{bmatrix}^T = \begin{bmatrix} \Delta V_x & \Delta V_y & \Delta V_z \end{bmatrix}^T. \tag{A4}
$$

For the kinematic equations, the previous relation becomes the final form.
Regarding the dynamic equations, aerodynamic terms can be written as:

$$
\boldsymbol{B}_I \begin{bmatrix} X^A \\ Y^A \\ Z^A \end{bmatrix} = -KV\boldsymbol{B}_I \begin{bmatrix} u \\ v \\ w \end{bmatrix} = -KV \begin{bmatrix} V_x \\ V_y \\ V_z \end{bmatrix}, \tag{A5}
$$

where $K = \frac{S}{2}\rho C_d$ depends on altitude $y_0$ and velocity $V$.

The linear form of the aerodynamic term becomes:

$$\Delta\left(\boldsymbol{B}_I\begin{bmatrix}X^A\\Y^A\\Z^A\end{bmatrix}\right) = -KV\begin{bmatrix}\Delta V_x\\\Delta V_y\\\Delta V_z\end{bmatrix} - K\begin{bmatrix}V_x\\V_y\\V_z\end{bmatrix}\Delta V - V\begin{bmatrix}V_x\\V_y\\V_z\end{bmatrix}\Delta K = -KV\begin{bmatrix}\Delta V_x\\\Delta V_y\\\Delta V_z\end{bmatrix} -$$

$$K\begin{bmatrix}V_x\\V_y\\V_z\end{bmatrix}\Delta V - V\begin{bmatrix}V_x\\V_y\\V_z\end{bmatrix}\frac{\partial K}{\partial V}\Delta V - V\begin{bmatrix}V_x\\V_y\\V_z\end{bmatrix}\frac{\partial K}{\partial y_0}\Delta y_0 = -KV\begin{bmatrix}\Delta V_x\\\Delta V_y\\\Delta V_z\end{bmatrix} - K\begin{bmatrix}V_x\\V_y\\V_z\end{bmatrix}\Delta V -$$

$$VK\frac{\partial C_d}{C_d\partial V}\begin{bmatrix}V_x\\V_y\\V_z\end{bmatrix}\Delta V - VK\left(\frac{\partial C_d}{C_d\partial y_0} + \frac{\partial\rho}{\rho\partial y_0}\right)\begin{bmatrix}V_x\\V_y\\V_z\end{bmatrix}\Delta y_0 = -KV\begin{bmatrix}\Delta V_x\\\Delta V_y\\\Delta V_z\end{bmatrix} -$$

$$K\left(1 + V\frac{\partial C_d}{C_d\partial V}\right)\begin{bmatrix}V_x\\V_y\\V_z\end{bmatrix}\Delta V - VK\left(\frac{\partial C_d}{C_d\partial y_0} + \frac{\partial\rho}{\rho\partial y_0}\right)\begin{bmatrix}V_x\\V_y\\V_z\end{bmatrix}\Delta y_0. \tag{A6}$$

The form of the aerodynamic term is:

$$\Delta\left(\boldsymbol{B}_I\begin{bmatrix}X^A\\Y^A\\Z^A\end{bmatrix}\right) = -KV\begin{bmatrix}\Delta V_x\\\Delta V_y\\\Delta V_z\end{bmatrix} - \frac{K}{V}\left(1 + V\frac{\partial C_d}{C_d\partial V}\right)\boldsymbol{V}_0\boldsymbol{V}_0^T\begin{bmatrix}\Delta V_x\\\Delta V_y\\\Delta V_z\end{bmatrix}$$

$$-VK\begin{bmatrix}V_x\\V_y\\V_z\end{bmatrix}\left(\frac{\partial C_d}{C_d\partial y_0} + \frac{\partial\rho}{\rho\partial y_0}\right)\Delta y_0, \tag{A7}$$

where:

$$\frac{\boldsymbol{V}_0\boldsymbol{V}_0^T}{V} = \frac{1}{V}\begin{bmatrix}V_x\\V_y\\V_z\end{bmatrix}\begin{bmatrix}V_x & V_y & V_z\end{bmatrix} = \frac{1}{V}\begin{bmatrix}V_x^2 & V_xV_y & V_xV_z\\V_yV_x & V_y^2 & V_yV_z\\V_zV_x & V_zV_y & V_z^2\end{bmatrix}, \tag{A8}$$

and:

$$\Delta K = \frac{S}{2}(C_d\Delta\rho + \Delta C_d\rho) = K\left(\frac{\Delta\rho}{\rho} + \frac{\Delta C_d}{C_d}\right) = K\left(\frac{1}{\rho}\frac{\partial\rho}{\partial y_0}\Delta y_0 + \frac{1}{C_d}\frac{\partial C_d}{\partial y_0}\Delta y_0 + \frac{1}{C_d}\frac{\partial C_d}{\partial V}\Delta V\right) =$$
$$bK\Delta y_0 + \frac{a-1}{V}K\Delta V. \tag{A9}$$

In order to evaluate the influence of altitude and velocity, based on the hypothesis of the spherical shape of the vehicle, for the considered velocity and altitude regime, through regression, the following approximation functions were obtained:

$$a = 1 + V\frac{\partial C_d}{C_d\partial V} \cong 1. - 0.5683\cdot V + 0.6409 \times 10^{-1}\cdot V^2 - 0.2332 \times 10^{-2}\cdot V^3;$$
$$b = \frac{\partial C_d}{C_d\partial y_0} + \frac{\partial\rho}{\rho\partial y_0} \cong 0.318 \times 10^{-4}, \tag{A10}$$

where:

$$\frac{\partial\rho}{\rho\partial y_0} = -0.96 \times 10^{-4}; \frac{\partial C_d}{C_d\partial y_0} = 1.278 \times 10^{-4}.$$

Next, we resume the development of the translational aerodynamic term (A7), which becomes:

$$\Delta\left(\boldsymbol{B}_I\begin{bmatrix}X^A\\Y^A\\Z^A\end{bmatrix}\right) = -KV\begin{bmatrix}\Delta V_x\\\Delta V_y\\\Delta V_z\end{bmatrix} - a\frac{K}{V}\begin{bmatrix}V_x^2 & V_xV_y & V_xV_z\\V_yV_x & V_y^2 & V_yV_z\\V_zV_x & V_zV_y & V_z^2\end{bmatrix}\begin{bmatrix}\Delta V_x\\\Delta V_y\\\Delta V_z\end{bmatrix} - bVK\begin{bmatrix}V_x\\V_y\\V_z\end{bmatrix}\Delta y_0$$

$$= -KV\left(I + \frac{a}{V^2}\begin{bmatrix}V_x^2 & V_xV_y & V_xV_z\\V_yV_x & V_y^2 & V_yV_z\\V_zV_x & V_zV_y & V_z^2\end{bmatrix}\right)\begin{bmatrix}\Delta V_x\\\Delta V_y\\\Delta V_z\end{bmatrix} - bKV\begin{bmatrix}V_x\\V_y\\V_z\end{bmatrix}\Delta y_0. \tag{A11}$$

Because in the basic movement $V_z = 0$, the translational aerodynamic term becomes:

$$
\Delta \left( \boldsymbol{B}_I \begin{bmatrix} X^A \\ Y^A \\ Z^A \end{bmatrix} \right) = -KV \left( I + \frac{a}{V^2} \begin{bmatrix} V_x^2 & V_x V_y & 0 \\ V_y V_x & V_y^2 & 0 \\ 0 & 0 & 0 \end{bmatrix} \right) \begin{bmatrix} \Delta V_x \\ \Delta V_y \\ \Delta V_z \end{bmatrix} - bKV \begin{bmatrix} V_x \\ V_y \\ 0 \end{bmatrix} \Delta y_0
$$

$$
= -KV \begin{bmatrix} 1 + \frac{aV_x^2}{V^2} & \frac{aV_x V_y}{V^2} & 0 \\ \frac{aV_x V_y}{V^2} & 1 + \frac{aV_y^2}{V^2} & 0 \\ 0 & 0 & 1 \end{bmatrix} \begin{bmatrix} \Delta V_x \\ \Delta V_y \\ \Delta V_z \end{bmatrix} - bKV \begin{bmatrix} V_x \\ V_y \\ 0 \end{bmatrix} \Delta y_0. \tag{A12}
$$

Returning to the dynamic equations of translation (A3), the linear form of the propulsive term is:

$$
\Delta \left( \boldsymbol{B}_I \begin{bmatrix} X^T \\ Y^T \\ Z^T \end{bmatrix} \right) = \boldsymbol{B}_I \begin{bmatrix} \Delta X^T \\ \Delta Y^T \\ \Delta Z^T \end{bmatrix} + \begin{bmatrix} \frac{\partial \boldsymbol{B}_I}{\partial q_1}; & \frac{\partial \boldsymbol{B}_I}{\partial q_2}; & \frac{\partial \boldsymbol{B}_I}{\partial q_3} \end{bmatrix} \begin{bmatrix} \boldsymbol{T} & & \\ & \boldsymbol{T} & \\ & & \boldsymbol{T} \end{bmatrix} \begin{bmatrix} \Delta q_1 \\ \Delta q_2 \\ \Delta q_3 \end{bmatrix}. \tag{A13}
$$

For the first term of the propulsive force development, we start from the nonlinear expression:

$$
\boldsymbol{T} = \begin{bmatrix} X^T \\ Y^T \\ Z^T \end{bmatrix} = \begin{bmatrix} \delta_T T_0 \cos\delta_m \cos\delta_n \\ -\delta_T T_0 \cos\delta_m \sin\delta_n \\ \delta_T T_0 \sin\delta_m \end{bmatrix}, \tag{A14}
$$

which can be expressed in the linear form:

$$
\begin{bmatrix} \Delta X^T \\ \Delta Y^T \\ \Delta Z^T \end{bmatrix} = T_0 \begin{bmatrix} \cos\delta_m \cos\delta_n & -\delta_T \cos\delta_m \sin\delta_n & -\delta_T \sin\delta_m \cos\delta_n \\ -\cos\delta_m \sin\delta_n & -\delta_T \cos\delta_m \cos\delta_n & \delta_T \sin\delta_m \sin\delta_n \\ \sin\delta_m & 0 & \delta_T \cos\delta_m \end{bmatrix} \begin{bmatrix} \Delta \delta_T \\ \Delta \delta_n \\ \Delta \delta_m \end{bmatrix}. \tag{A15}
$$

Because in the basic movement, the yaw command is null $\delta_m = 0$, we will obtain:

$$
\begin{bmatrix} \Delta X^T \\ \Delta Y^T \\ \Delta Z^T \end{bmatrix} = T_0 \begin{bmatrix} \cos\delta_n & -\delta_T \sin\delta_n & 0 \\ -\sin\delta_n & -\delta_T \cos\delta_n & 0 \\ 0 & 0 & \delta_T \end{bmatrix} \begin{bmatrix} \Delta \delta_T \\ \Delta \delta_n \\ \Delta \delta_m \end{bmatrix}. \tag{A16}
$$

For the second term of relation (A13), relying on the inverse rotation matrix:

$$
\boldsymbol{B}_I = \begin{bmatrix} q_4^2 + q_1^2 - q_2^2 - q_3^2 & 2(q_1 q_2 - q_3 q_4) & 2(q_3 q_1 + q_2 q_4) \\ 2(q_1 q_2 + q_3 q_4) & q_4^2 + q_2^2 - q_3^2 - q_1^2 & 2(q_2 q_3 - q_4 q_1) \\ 2(q_3 q_1 - q_2 q_4) & 2(q_2 q_3 + q_4 q_1) & q_4^2 + q_3^2 - q_1^2 - q_2^2 \end{bmatrix}, \tag{A17}
$$

the derivatives with respect to the quaternion components are:

$$
\frac{\partial \boldsymbol{B}_I}{\partial q_1} = -\frac{q_1}{q_4} \frac{\partial \boldsymbol{B}_I}{\partial q_4} + 2 \begin{bmatrix} q_1 & q_2 & q_3 \\ q_2 & -q_1 & -q_4 \\ q_3 & q_4 & -q_1 \end{bmatrix}; \frac{\partial \boldsymbol{B}_I}{\partial q_2} = -\frac{q_2}{q_4} \frac{\partial \boldsymbol{B}_I}{\partial q_4} + 2 \begin{bmatrix} -q_2 & q_1 & q_4 \\ q_1 & q_2 & q_3 \\ -q_4 & q_3 & -q_2 \end{bmatrix};
$$

$$
\frac{\partial \boldsymbol{B}_I}{\partial q_3} = -\frac{q_3}{q_4} \frac{\partial \boldsymbol{B}_I}{\partial q_4} + 2 \begin{bmatrix} -q_3 & -q_4 & q_1 \\ q_4 & -q_3 & q_2 \\ q_1 & q_2 & q_3 \end{bmatrix}; \frac{\partial \boldsymbol{B}_I}{\partial q_4} = 2 \begin{bmatrix} q_4 & -q_3 & q_2 \\ q_3 & q_4 & -q_1 \\ -q_2 & q_1 & q_4 \end{bmatrix}. \tag{A18}
$$

In the basic movement, we have: $q_1 = 0$; $q_2 = 0$; $\delta_m = 0$, from where:

$$\boldsymbol{T} = \begin{bmatrix} X^T \\ Y^T \\ Z^T \end{bmatrix} = \begin{bmatrix} \delta_T T_0 cos\delta_n \\ -\delta_T T_0 sin\delta_n \\ 0 \end{bmatrix};$$

$$\frac{\partial \boldsymbol{B}_I}{\partial q_1} = 2 \begin{bmatrix} 0 & 0 & q_3 \\ 0 & 0 & -q_4 \\ q_3 & q_4 & 0 \end{bmatrix}; \frac{\partial \boldsymbol{B}_I}{\partial q_2} = 2 \begin{bmatrix} 0 & 0 & q_4 \\ 0 & 0 & q_3 \\ -q_4 & q_3 & 0 \end{bmatrix}; \frac{\partial \boldsymbol{B}_I}{\partial q_3} = -\frac{q_3}{q_4}\frac{\partial \boldsymbol{B}_I}{\partial q_4} +$$

$$2 \begin{bmatrix} -q_3 & -q_4 & 0 \\ q_4 & -q_3 & 0 \\ 0 & 0 & q_3 \end{bmatrix} =$$

$$-\frac{2q_3}{q_4}\begin{bmatrix} q_4 & -q_3 & 0 \\ q_3 & q_4 & 0 \\ 0 & 0 & q_4 \end{bmatrix} + 2\begin{bmatrix} -q_3 & -q_4 & 0 \\ q_4 & -q_3 & 0 \\ 0 & 0 & q_3 \end{bmatrix} = \frac{2}{q_4}\begin{bmatrix} -q_3q_4 & -q_4q_4 & 0 \\ q_4q_4 & -q_3q_4 & 0 \\ 0 & 0 & q_3q_4 \end{bmatrix} -$$

$$\frac{2}{q_4}\begin{bmatrix} q_3q_4 & -q_3q_3 & 0 \\ q_3q_3 & q_4q_3 & 0 \\ 0 & 0 & q_4q_3 \end{bmatrix} = \frac{2}{q_4}\begin{bmatrix} -2q_3q_4 & q_3q_3 - q_4q_4 & 0 \\ q_4q_4 - q_3q_3 & -2q_3q_4 & 0 \\ 0 & 0 & 0 \end{bmatrix};$$

$$\frac{\partial \boldsymbol{B}_I}{\partial q_3} = \frac{2}{q_4}\begin{bmatrix} -2q_3q_4 & 2q_3^2 - 1 & 0 \\ 1 - 2q_3^2 & -2q_3q_4 & 0 \\ 0 & 0 & 0 \end{bmatrix}; \boldsymbol{B}_I = \begin{bmatrix} 1 - 2q_3^2 & -2q_3q_4 & 0 \\ 2q_3q_4 & 1 - 2q_3^2 & 0 \\ 0 & 0 & 1 \end{bmatrix}. \quad \text{(A19)}$$

The first propulsive term from relation (A13) is:

$$\boldsymbol{B}_I\begin{bmatrix} \Delta X^T \\ \Delta Y^T \\ \Delta Z^T \end{bmatrix} = T_0\begin{bmatrix} (1 - 2q_3^2) & -2q_3q_4 & 0 \\ 2q_3q_4 & (1 - 2q_3^2) & 0 \\ 0 & 0 & 1 \end{bmatrix}\begin{bmatrix} cos\delta_n & -\delta_T sin\delta_n & 0 \\ -sin\delta_n & -\delta_T cos\delta_n & 0 \\ 0 & 0 & \delta_T \end{bmatrix}\begin{bmatrix} \Delta\delta_T \\ \Delta\delta_n \\ \Delta\delta_m \end{bmatrix} =$$

$$T_0\begin{bmatrix} 2q_3q_4 sin\delta_n + (1 - 2q_3^2)cos\delta_n & 2\delta_T q_3q_4 cos\delta_n - \delta_T(1 - 2q_3^2)sin\delta_n & 0 \\ 2q_3q_4 cos\delta_n - (1 - 2q_3^2)sin\delta_n & -2\delta_T q_3q_4 sin\delta_n - \delta_T(1 - 2q_3^2)cos\delta_n & 0 \\ 0 & 0 & \delta_T \end{bmatrix}\begin{bmatrix} \Delta\delta_T \\ \Delta\delta_n \\ \Delta\delta_m \end{bmatrix}, \quad \text{(A20)}$$

and the second one becomes:

$$\begin{bmatrix} \frac{\partial \boldsymbol{B}_I}{\partial q_1}; & \frac{\partial \boldsymbol{B}_I}{\partial q_2}; & \frac{\partial \boldsymbol{B}_I}{\partial q_3} \end{bmatrix}\begin{bmatrix} \boldsymbol{T} & & \\ & \boldsymbol{T} & \\ & & \boldsymbol{T} \end{bmatrix}\begin{bmatrix} \Delta q_1 \\ \Delta q_2 \\ \Delta q_3 \end{bmatrix} =$$

$$\frac{2}{q_4}\begin{bmatrix} 0 & 0 & -2q_3q_4 X^T + (2q_3^2 - 1)Y^T \\ 0 & 0 & (1 - 2q_3^2)X^T - 2q_3q_4 Y^T \\ q_3q_4 X^T + q_4^2 Y^T & -q_4^2 X^T + q_3q_4 Y^T & 0 \end{bmatrix}\begin{bmatrix} \Delta q_1 \\ \Delta q_2 \\ \Delta q_3 \end{bmatrix} = \quad \text{(A21)}$$

$$\frac{2\delta_T T_0}{q_4}\begin{bmatrix} 0 & 0 & P_{13} \\ 0 & 0 & P_{23} \\ P_{31} & P_{32} & 0 \end{bmatrix}\begin{bmatrix} \Delta q_1 \\ \Delta q_2 \\ \Delta q_3 \end{bmatrix},$$

where:

$$P_{31} = q_3q_4 cos\delta_n - q_4^2 sin\delta_n, P_{32} = -q_4^2 cos\delta_n - q_3q_4 sin\delta_n, P_{13} = -2q_3q_4 cos\delta_n + (1 - 2q_3^2)sin\delta_n, P_{23} = (1 - 2q_3^2)cos\delta_n + 2q_3q_4 sin\delta_n.$$

Cumulating the two terms, the linear form of the propulsive term from the translation Equation (A13) is obtained:

$$\Delta\left(\boldsymbol{B}_I\begin{bmatrix} X^T \\ Y^T \\ Z^T \end{bmatrix}\right) = \frac{2\delta_T T_0}{q_4}\begin{bmatrix} 0 & 0 & P_{13} \\ 0 & 0 & P_{23} \\ P_{31} & P_{32} & 0 \end{bmatrix}\begin{bmatrix} \Delta q_1 \\ \Delta q_2 \\ \Delta q_3 \end{bmatrix} +$$

$$T_0\begin{bmatrix} 2q_3q_4 sin\delta_n + (1 - 2q_3^2)cos\delta_n & 2\delta_T q_3q_4 cos\delta_n - \delta_T(1 - 2q_3^2)sin\delta_n & 0 \\ 2q_3q_4 cos\delta_n - (1 - 2q_3^2)sin\delta_n & -2\delta_T q_3q_4 sin\delta_n - \delta_T(1 - 2q_3^2)cos\delta_n & 0 \\ 0 & 0 & \delta_T \end{bmatrix}\begin{bmatrix} \Delta\delta_T \\ \Delta\delta_n \\ \Delta\delta_m \end{bmatrix}. \quad \text{(A22)}$$

*A.2. Rotational Movement*

In this section, we will obtain the linear form of the rotational equations in the body frame:

The nonlinear rotational equations are:

The dynamic equations:

$$
\begin{bmatrix} \dot{p} \\ \dot{q} \\ \dot{r} \end{bmatrix} = J^{-1} \begin{bmatrix} L \\ M \\ N \end{bmatrix} - J^{-1} A_\Omega J \begin{bmatrix} p \\ q \\ r \end{bmatrix}. \tag{A23}
$$

The kinematic equations:

$$
\begin{bmatrix} \dot{q_1} \\ \dot{q_2} \\ \dot{q_3} \\ \dot{q_4} \end{bmatrix} = \begin{bmatrix} q_4 & -q_3 & q_2 \\ q_3 & q_4 & -q_1 \\ -q_2 & q_1 & q_4 \\ -q_1 & -q_2 & -q_3 \end{bmatrix} \begin{bmatrix} p \\ q \\ r \end{bmatrix}. \tag{A24}
$$

If we consider as a basic movement the translation in the vertical plane, we have: $q_1 = 0$; $q_2 = 0$; $q_4 = \pm\sqrt{1 - q_3^2}$, with zero angular velocity: $p = 0$, $q = 0$, $r = 0$, the linear form of the rotational equations become:

The dynamic equations:

$$
\begin{bmatrix} \Delta\dot{p} \\ \Delta\dot{q} \\ \Delta\dot{r} \end{bmatrix} = \begin{bmatrix} 1/A & 0 & 0 \\ 0 & 1/B & 0 \\ 0 & 0 & 1/C \end{bmatrix} \begin{bmatrix} \Delta L \\ \Delta M \\ \Delta N \end{bmatrix}. \tag{A25}
$$

The kinematic equations

$$
\begin{bmatrix} \Delta\dot{q_1} \\ \Delta\dot{q_2} \\ \Delta\dot{q_3} \end{bmatrix} = \frac{1}{2} \begin{bmatrix} q_4 & -q_3 & 0 \\ q_3 & q_4 & 0 \\ 0 & 0 & q_4 \end{bmatrix} \begin{bmatrix} \Delta p \\ \Delta q \\ \Delta r \end{bmatrix}. \tag{A26}
$$

For the kinematic equations, the previous relation becomes the final form.

Regarding the dynamic equations, linear forms of the applied moments must be expressed.

For the development of the aerodynamic moment, we start from the aerodynamic force in the body frame, which can be expressed as follows:

$$
\begin{bmatrix} X^A \\ Y^A \\ Z^A \end{bmatrix} = -KV \begin{bmatrix} u \\ v \\ w \end{bmatrix}. \tag{A27}
$$

Expanding linearly, we obtain:

$$
\begin{bmatrix} \Delta X^A \\ \Delta Y^A \\ \Delta Z^A \end{bmatrix} = -KV \begin{bmatrix} \Delta u \\ \Delta v \\ \Delta w \end{bmatrix} - K \begin{bmatrix} u \\ v \\ w \end{bmatrix} \Delta V - V \begin{bmatrix} u \\ v \\ w \end{bmatrix} \Delta K. \tag{A28}
$$

The three terms can be developed separately:

$$
KV \begin{bmatrix} \Delta u \\ \Delta v \\ \Delta w \end{bmatrix} = KV \boldsymbol{A}_I \begin{bmatrix} \Delta V_x \\ \Delta V_y \\ \Delta V_z \end{bmatrix} + KV \begin{bmatrix} \frac{\partial \boldsymbol{A}_I}{\partial q_1}; & \frac{\partial \boldsymbol{A}_I}{\partial q_2}; & \frac{\partial \boldsymbol{A}_I}{\partial q_3} \end{bmatrix} \begin{bmatrix} \boldsymbol{V}_0 & & \\ & \boldsymbol{V}_0 & \\ & & \boldsymbol{V}_0 \end{bmatrix} \begin{bmatrix} \Delta q_1 \\ \Delta q_2 \\ \Delta q_3 \end{bmatrix};
$$

$$
K \begin{bmatrix} u \\ v \\ w \end{bmatrix} \Delta V = \frac{K}{V} \boldsymbol{A}_I \boldsymbol{V}_0 \boldsymbol{V}_0^T \begin{bmatrix} \Delta V_x \\ \Delta V_y \\ \Delta V_z \end{bmatrix};
$$

$$
V \begin{bmatrix} u \\ v \\ w \end{bmatrix} \Delta K = bVK\boldsymbol{A}_I \begin{bmatrix} V_x \\ V_y \\ V_z \end{bmatrix} \Delta y_0 + (a-1)K\boldsymbol{A}_I \begin{bmatrix} V_x \\ V_y \\ V_z \end{bmatrix} \Delta V = bVK\boldsymbol{A}_I \begin{bmatrix} V_x \\ V_y \\ V_z \end{bmatrix} \Delta y_0 +
$$

$$
\frac{(a-1)}{V} K \boldsymbol{A}_I \boldsymbol{V}_0 \boldsymbol{V}_0^T \begin{bmatrix} \Delta V_x \\ \Delta V_y \\ \Delta V_z \end{bmatrix}, \tag{A29}
$$

thus resulting:

$$
\begin{bmatrix} \Delta X^A \\ \Delta Y^A \\ \Delta Z^A \end{bmatrix} = -KV\boldsymbol{A}_I \begin{bmatrix} \Delta V_x \\ \Delta V_y \\ \Delta V_z \end{bmatrix} - KV \begin{bmatrix} \frac{\partial \boldsymbol{A}_I}{\partial q_1}; & \frac{\partial \boldsymbol{A}_I}{\partial q_2}; & \frac{\partial \boldsymbol{A}_I}{\partial q_3} \end{bmatrix} \begin{bmatrix} \boldsymbol{V}_0 & & \\ & \boldsymbol{V}_0 & \\ & & \boldsymbol{V}_0 \end{bmatrix} \begin{bmatrix} \Delta q_1 \\ \Delta q_2 \\ \Delta q_3 \end{bmatrix} -
$$

$$
aK\boldsymbol{A}_I \frac{\boldsymbol{V}_0 \boldsymbol{V}_0^T}{V} \begin{bmatrix} \Delta V_x \\ \Delta V_y \\ \Delta V_z \end{bmatrix} - bVK\boldsymbol{A}_I \begin{bmatrix} V_x \\ V_y \\ V_z \end{bmatrix} \Delta y_0, \tag{A30}
$$

or yet

$$
\begin{bmatrix} \Delta X^A \\ \Delta Y^A \\ \Delta Z^A \end{bmatrix} = -KV\boldsymbol{A}_I \left( \boldsymbol{I} + \frac{a\boldsymbol{V}_0 \boldsymbol{V}_0^T}{V^2} \right) \begin{bmatrix} \Delta V_x \\ \Delta V_y \\ \Delta V_z \end{bmatrix} - KV \begin{bmatrix} \frac{\partial \boldsymbol{A}_I}{\partial q_1}; & \frac{\partial \boldsymbol{A}_I}{\partial q_2}; & \frac{\partial \boldsymbol{A}_I}{\partial q_3} \end{bmatrix} \begin{bmatrix} \boldsymbol{V}_0 & & \\ & \boldsymbol{V}_0 & \\ & & \boldsymbol{V}_0 \end{bmatrix} \begin{bmatrix} \Delta q_1 \\ \Delta q_2 \\ \Delta q_3 \end{bmatrix} -
$$

$$
bVK\boldsymbol{A}_I \begin{bmatrix} V_x \\ V_y \\ V_z \end{bmatrix} \Delta y_0, \tag{A31}
$$

where

$$
\frac{\partial \boldsymbol{A}_I}{\partial q_1} = -\frac{q_1}{q_4} \frac{\partial \boldsymbol{A}_I}{\partial q_4} + 2 \begin{bmatrix} q_1 & q_2 & q_3 \\ q_2 & -q_1 & q_4 \\ q_3 & -q_4 & -q_1 \end{bmatrix}; \quad \frac{\partial \boldsymbol{A}_I}{\partial q_2} = -\frac{q_2}{q_4} \frac{\partial \boldsymbol{A}_I}{\partial q_4} + 2 \begin{bmatrix} -q_2 & q_1 & -q_4 \\ q_1 & q_2 & q_3 \\ q_4 & q_3 & -q_2 \end{bmatrix};
$$

$$
\frac{\partial \boldsymbol{A}_I}{\partial q_3} = -\frac{q_3}{q_4} \frac{\partial \boldsymbol{A}_I}{\partial q_4} + 2 \begin{bmatrix} -q_3 & q_4 & q_1 \\ -q_4 & -q_3 & q_2 \\ q_1 & q_2 & q_3 \end{bmatrix}; \quad \frac{\partial \boldsymbol{A}_I}{\partial q_4} = 2 \begin{bmatrix} q_4 & q_3 & -q_2 \\ -q_3 & q_4 & q_1 \\ q_2 & -q_1 & q_4 \end{bmatrix}. \tag{A32}
$$

In the basic movement case, $q_1 = 0$; $q_2 = 0$; $V_z = 0$, from where:

$$
\boldsymbol{V}_0 = \begin{bmatrix} V_x \\ V_y \\ 0 \end{bmatrix};
$$

$$
\frac{\boldsymbol{V}_0 \boldsymbol{V}_0^T}{V^2} = \frac{1}{V^2} \begin{bmatrix} V_x^2 & V_x V_y & 0 \\ V_y V_x & V_y^2 & 0 \\ 0 & 0 & 0 \end{bmatrix};
$$

$$
\frac{\partial \boldsymbol{A}_I}{\partial q_1} = 2 \begin{bmatrix} 0 & 0 & q_3 \\ 0 & 0 & q_4 \\ q_3 & -q_4 & 0 \end{bmatrix}; \quad \frac{\partial \boldsymbol{A}_I}{\partial q_2} = 2 \begin{bmatrix} 0 & 0 & -q_4 \\ 0 & 0 & q_3 \\ q_4 & q_3 & 0 \end{bmatrix};
$$

$$
\frac{\partial \boldsymbol{A}_I}{\partial q_3} = \frac{2}{q_4} \begin{bmatrix} -2q_3 q_4 & 1 - 2q_3^2 & 0 \\ 2q_3^2 - 1 & -2q_3 q_4 & 0 \\ 0 & 0 & 0 \end{bmatrix}; \quad \boldsymbol{A}_I = \begin{bmatrix} 1 - 2q_3^2 & 2q_3 q_4 & 0 \\ -2q_3 q_4 & 1 - 2q_3^2 & 0 \\ 0 & 0 & 1 \end{bmatrix}. \tag{A33}
$$

In this case, after successive computations, the relation (A31) becomes:

$$
\begin{bmatrix} \Delta X^A \\ \Delta Y^A \\ \Delta Z^A \end{bmatrix} = -KV \begin{bmatrix} Q_{11} & Q_{12} & 0 \\ Q_{21} & Q_{22} & 0 \\ 0 & 0 & 1 \end{bmatrix} \begin{bmatrix} \Delta V_x \\ \Delta V_y \\ \Delta V_z \end{bmatrix} - 2\frac{KV}{q_4} \begin{bmatrix} 0 & 0 & R_{13} \\ 0 & 0 & R_{23} \\ R_{31} & R_{32} & 0 \end{bmatrix} \begin{bmatrix} \Delta q_1 \\ \Delta q_2 \\ \Delta q_3 \end{bmatrix}
$$
$$
+ aVK \begin{bmatrix} \left(1 - 2q_3^2\right)V_x + 2q_3q_4V_y \\ -2q_3q_4V_x + \left(1 - 2q_3^2\right)V_y \\ 0 \end{bmatrix} \Delta y_0,
\tag{A34}
$$

where

$$
Q_{11} = 2q_3q_4\frac{aV_xV_y}{V^2} + \left(1 - 2q_3^2\right)\left(1 + \frac{aV_x^2}{V^2}\right), \quad Q_{12} = 2q_3q_4\left(1 + \frac{aV_y^2}{V^2}\right) +
$$
$$
\left(1 - 2q_3^2\right)\frac{aV_xV_y}{V^2}, \quad Q_{21} = -2q_3q_4\left(1 + \frac{aV_x^2}{V^2}\right) + \left(1 - 2q_3^2\right)\frac{aV_xV_y}{V^2}, \quad Q_{22} = -2q_3q_4\frac{aV_xV_y}{V^2} +
$$
$$
\left(1 - 2q_3^2\right)\left(1 + \frac{aV_y^2}{V^2}\right).
$$
$$
R_{13} = -2q_3q_4V_x + \left(1 - 2q_3^2\right)V_y, \quad R_{23} = -\left(1 - 2q_3^2\right)V_x - 2q_3q_4V_y, \quad R_{31} = \left(q_3V_x - q_4V_y\right)q_4,
$$
$$
R_{32} = \left(q_4V_x + q_3V_y\right)q_4.
$$

Regarding the aerodynamic moments, for the configuration considered, the linear form is:

$$
\Delta L^A = 0; \; \Delta M^A = -x_F \Delta Z^A \; ; \; \Delta N^A = x_F \Delta Y^A.
\tag{A35}
$$

Considering the development of the aerodynamic force presented previously (A34), we obtain:

$$
\begin{bmatrix} \Delta L^A \\ \Delta M^A \\ \Delta N^A \end{bmatrix} = -KVx_F \begin{bmatrix} 0 & 0 & 0 \\ 0 & 0 & -1 \\ S_{31} & S_{32} & 0 \end{bmatrix} \begin{bmatrix} \Delta V_x \\ \Delta V_y \\ \Delta V_z \end{bmatrix}
$$
$$
-2\frac{KVx_F}{q_4} \begin{bmatrix} 0 & 0 & 0 \\ -\left(q_3V_x - q_4V_y\right)q_4 & -\left(q_4V_x + q_3V_y\right)q_4 & 0 \\ 0 & 0 & -\left(1 - 2q_3^2\right)V_x - 2q_3q_4V_y \end{bmatrix} \begin{bmatrix} \Delta q_1 \\ \Delta q_2 \\ \Delta q_3 \end{bmatrix}
$$
$$
-bKx_FV \begin{bmatrix} 0 \\ 0 \\ -2q_3q_4V_x + \left(1 - 2q_3^2\right)V_y \end{bmatrix} \Delta y_0,
\tag{A36}
$$

where

$$
S_{31} = -2q_3q_4\left(1 + \frac{aV_x^2}{V^2}\right) + \left(1 - 2q_3^2\right)\frac{aV_xV_y}{V^2}, \quad S_{32} = -2q_3q_4\frac{aV_xV_y}{V^2} + \left(1 - 2q_3^2\right)\left(1 + \frac{aV_y^2}{V^2}\right).
$$

For the propulsive moments of (A25), we resume the linear form of the translation terms (A16):

$$
\begin{bmatrix} \Delta X^T \\ \Delta Y^T \\ \Delta Z^T \end{bmatrix} = T_0 \begin{bmatrix} \cos\delta_n & -\delta_T\sin\delta_n & 0 \\ -\sin\delta_n & -\delta_T\cos\delta_n & 0 \\ 0 & 0 & \delta_T \end{bmatrix} \begin{bmatrix} \Delta\delta_T \\ \Delta\delta_n \\ \Delta\delta_m \end{bmatrix},
\tag{A37}
$$

and we consider that the command propulsive moments are:

$$
\Delta M^T = -x_T \Delta Z^T = -x_T T_0 \delta_T \Delta\delta_m;
$$
$$
\Delta N^T = x_T \Delta Y^T = -x_T T_0 \sin\delta_n \Delta\delta_T - x_T T_0 \delta_T\cos\delta_n \Delta\delta_n.
\tag{A38}
$$

Regarding the roll moment command, it is considered as:

$$
\Delta L^T = Rd\Delta\delta_l,
\tag{A39}
$$

where $\delta_l$ is the equivalent roll deflection.

The linear form of the propulsive moment term becomes:

$$
\begin{bmatrix} \Delta L^T \\ \Delta M^T \\ \Delta N^T \end{bmatrix} = \begin{bmatrix} 0 & 0 & Rd & 0 \\ 0 & 0 & 0 & -x_T T_0 \delta_T \\ -x_T T_0 sin\delta_n & -x_T T_0 \delta_T cos\delta_n & 0 & 0 \end{bmatrix} \begin{bmatrix} \Delta\delta_T \\ \Delta\delta_n \\ \Delta\delta_l \\ \Delta\delta_m \end{bmatrix}, \qquad (A40)
$$

Thus, obtaining the linear form for the dynamic equations of rotation in the body frame.

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
