# Peer review of "Performance Evaluation for Launcher Testing Vehicle"

_aerospace, doi:10.3390/aerospace9090504_

Round 1

Reviewer 1 Report

The paper is well-written in general. However, there are too many paragraphs and sections. They can be combined and better structured. Also, please clarify the state-of-the-art contribution in the introduction.

Author Response

Thank you very much for the comments concerning our manuscript. Please see the attachment

Reviewer 2 Report

this paper present a interesting topic, theory, explanation and development of the model are clear. However, you omitted the conclusions. I recommend reviewing it intensively and correcting minor omissions or errors.

Author Response

(The authors gave the same response as above.)

Reviewer 3 Report

- The paper is well presented but it is very similar to a research report. 
 - The conclusions are absent, there are no difference with the other models in literature. 

- The example made are very far to a real LV (i.e. We will considerate a hypothetical LTV with initial mass 100 kg, from which prope lant is 70 kg, equipped with liquid rocket engine with specific impulse ???=230[?], spe cific for regular liquid engine (Kerosene + LOX), with maximum thrust ?0=1200 [?] and reactive force of one RCS element ?=5 [?] .) and so it is difficult to understand if the models presented are reliable or not. The subparts of the models could be, in principle, compared with some similar examples in literature.  As example in the highlighted part it could be used something from literature for what concerns thruster performances. 

The manuscript shall be improved in all its part and, at that point, it could be considered for publication.

Author Response

Thank you very much for the comments concerning our manuscript.Please see the attachment.
